**EMBO** *reports*

# MIA40 suppresses cell death induced by apoptosis-inducing factor 1

Ben Hur Marins Mussulini [1,2], Klaudia K Maruszczak [2], Piotr Draczkowski [3,4], Mayra A Borrero-Landazabal [2], Selvaraj Ayyamperumal [2], Artur Wnorowski[5], Michal Wasilewski [1,2] & Agnieszka Chacinska [1,2]✉

## Abstract

Mitochondria harbor respiratory complexes that perform oxidative phosphorylation. Complex I is the first enzyme of the respiratory chain that oxidizes NADH. A dysfunction in complex I can result in higher cellular levels of NADH, which in turn strengthens the interaction between apoptosis-inducing factor 1 (AIFM1) and Mitochondrial intermembrane space import and assembly protein 40 (MIA40) in the mitochondrial intermembrane space. We investigated whether MIA40 modulates the activity of AIFM1 upon increased NADH/NAD+ balance. We found that in model cells characterized by an increase in NADH the AIFM1-MIA40 interaction is strengthened and these cells demonstrate resistance to AIFM1-induced cell death. Either silencing of MIA40, rescue of complex I, or depletion of NADH through the expression of yeast NADH-ubiquinone oxidoreductase-2 sensitized NDUFA13-KO cells to AIFM1-induced cell death. These findings indicate that the complex of MIA40 and AIFM1 suppresses AIFM1-induced cell death in a NADH-dependent manner. This study identifies an effector complex involved in regulating the programmed cell death that accommodates the metabolic changes in the cell and provides a molecular explanation for AIFM1-mediated chemoresistance of cancer cells.

**Keywords** Programmed Cell Death; Mitochondria; Metabolism; Protein Import; Cancer
**Subject Categories** Autophagy & Cell Death; Membranes & Trafficking; Metabolism

## Introduction

Mitochondria are multifunctional semi-autonomous organelles that are involved in cell metabolism and cell death. They are confined by two membranes that define two intramitochondrial compartments: intermembrane space (IMS) and matrix. The majority of proteins that are required for mitochondrial biogenesis are encoded in the nuclear DNA and imported into the organelle via complex import pathways (Pfanner et al, 2019; Wasilewski et al, 2017). The mitochondrial intermembrane space import and assembly (MIA) pathway, which is responsible for the translocation of cysteine-rich proteins to the IMS, relies on MIA40/CHCHD4 oxidoreductase (Barchiesi et al, 2015; Chacinska et al, 2008; Chacinska et al, 2004; Mohanraj et al, 2019). MIA40 canonical substrates contain specific motifs in their sequence, namely twin $CX_3C$ or twin $CX_9C$. MIA40 mediates the oxidative folding of newly imported proteins, such as the translocase of the inner mitochondrial membrane 8A (TIMM8A) and cytochrome C oxidase 6B (COX6B) (Banci et al, 2009; Bourens et al, 2012; Fischer et al, 2013; Koehler, 2004; Milenkovic et al, 2009). MIA40 does so by introducing intramolecular disulfide bonds via the redox activity of its CPC domain (Cys53 and Cys55) and a hydrophobic cleft (Banci et al, 2009; Habich et al, 2019b; Peleh et al, 2016). Non-canonical substrates can also be imported by MIA40, such as the translocase of the inner mitochondrial membrane 22 (Tim22) in yeast or mitochondrial calcium uptake 1 (MICU1), purinic/apyrimidinic endonuclease (APE1), cytochrome C oxidase and assembly factor 7 (COA7) in metazoa (Barchiesi et al, 2015; Mohanraj et al, 2019; Petrungaro et al, 2015; Wrobel et al, 2013; Zhuang et al, 2013).

MIA40 itself is transported to the IMS by the association of its unstructured N-terminal region with apoptosis-inducing factor 1 (AIFM1), which serves as an import receptor of MIA40 (Hangen et al, 2015; Meyer et al, 2015; Petrungaro et al, 2015). In the IMS, depending on the tissue, MIA40 can exist in a free-floating form or, in presence of β-nicotinamide adenine dinucleotide (NADH), in a stable complex with AIFM1 (Hangen et al, 2015; Salscheider et al, 2022). AIFM1 is located in the inner mitochondrial membrane but upon loss of outer mitochondrial membrane integrity, AIFM1 translocates to the nucleus and promotes cell death by DNA fragmentation in a caspase-independent chromatin condensation manner (Candé et al, 2004; Nakao et al, 2015; Ravagnan et al, 2001; Shelar et al, 2015; Susin et al, 1999; Zhu et al, 2007). The absence of AIFM1 results in electron transport chain (ETC) defects that are connected to the downregulation of complexes I, III, and IV (Bénit et al, 2008; Brown et al, 2006; Pospisilik et al, 2007). Complex I (NADH: ubiquinone oxidoreductase) is a boot-shaped assembly

[1]ReMedy International Research Agenda Unit, IMol Polish Academy of Sciences, Warsaw, Poland. [2]IMol Polish Academy of Sciences, Warsaw, Poland. [3]National Bioinformatics Infrastructure Sweden, SciLifeLab, Solna, Sweden. [4]Department of Synthesis and Chemical Technology of Pharmaceutical Substances, Medical University of Lublin, Lublin, Poland. [5]Department of Biopharmacy, Medical University of Lublin, Lublin, Poland. ✉E-mail: a.chacinska@imol.institute

that is composed of 31 accessory subunits surrounding 14 core subunits (Stroud et al, 2016). Complex I oxidizes NADH, regenerating NAD$^+$ levels in the mitochondrial matrix (Hirst, 2013), and regulates cellular NADH/NAD$^+$ ratios via multiple mitochondrial substrate shuttling systems (Xiao et al, 2018). Both mitochondrial and nuclear gene expression contribute to complex I assembly, and mutations of the genetic material of either can result in pathologies, such as cancer (Yang et al, 2022; Iommarini et al, 2013). Impairments in complex I may result in increased cellular NADH/NAD$^+$ balance (Yang et al, 2022) e.g., during breast cancer progression (Santidrian et al, 2013). Cancer has a plethora of mechanisms to avoid cell death, among them AIFM1-dependent chemoresistance (Weiss et al, 2022; Yang et al, 2008; Urbano et al, 2005). Interestingly, although the AIFM1-MIA40 interaction is NADH-dependent (Hangen et al, 2015; Salscheider et al, 2022), little is known about the modulation of AIFM1 activity by MIA40.

Here, we found an increased AIFM1-MIA40 interaction and resistance to AIFM1-induced cell death in a complex I deficient cellular model in which NADH:ubiquinone oxidoreductase subunit A13 was knocked out (NDUFA13-KO). We showed that the deletion of NDUFA13 resulted in increased NADH/NAD$^+$ balance. Subsequently, we investigated a role of MIA40 in the modulation of AIFM1 cell death activity. We showed that the silencing of MIA40 sensitized NDUFA13-KO cells to AIFM1-induced cell death. Lastly, we showed that NADH depletion led to a decrease in the AIFM1-MIA40 interaction and recovered AIFM1-induced cell death in NDUFA13-KO cells. Overall, our findings demonstrate that MIA40 modulates the activity of AIFM1 in a NADH-dependent manner, revealing a new regulation of cell death, which responds to changes in metabolic status of NADH.

# Results

## Deletion of a complex I subunit, NDUFA13, as a model for increased NADH/NAD$^+$ balance

To elucidate whether MIA40 modulates AIFM1 cell death activity upon increased NADH/NAD$^+$ balance, we took advantage of a mitochondrial complex I-deficient human cell library (Stroud et al, 2016). We selected cell lines that were deleted for complex I accessory subunits known to have no major, partial effect or complete effect on inhibition of complex I activity: knockouts of NDUFA7, NDUFS6, NDUFV3, NDUFA13, NDUFA5, NDUFS5, NDUFB4, and NDUFB7 (Stroud et al, 2016). For all models, we induced cell death by blocking caspase activity prior to treatment with staurosporine, which triggers AIFM1-induced cell death (Susin et al, 1999). Subsequently, overall cellular metabolic viability and cellular survival were measured (Fig. EV1A,B). While several knockout cell lines were equally sensitive to the induction of cell death as the control HEK293T cells, some cell lines reacted differently. In particular, the NDUFV3- and NDUFA13-KO cells exhibited increased resistance to cell death, whereas NDUFA5- and NDUFB4-KO cells were more sensitive to AIFM1-induced cell death. The observed cell death phenotypes were further corroborated by flow cytometry (Fig. 1A,B). We then verified how the resistance or sensitivity to cell death in these cells correlated to their energetic state. To this end, we measured the activity of complex I and the NADH/NAD$^+$ balance in the selected cells. We found that

the NDUFV3-KO cells exhibited a 20% decrease in the complex I activity compared to the control HEK293T cells, whereas the NDUFA13-, NDUFA5-, and NDUFB4-KO cells displayed a more severe decrease in the complex I activity of ~90%. (Fig. EV1C). Therefore, we concluded that the degree of complex I impairment did not correlate with the sensitivity or resistance to AIFM1-induced cell death in these cells. Finally, we measured the NADH/NAD$^+$ balance and found that the cells that were resistant to AIFM1-induced cell death, NDUFV3- and NDUFA13-KO, had also an increased NADH/NAD$^+$ ratio (Fig. EV1D). Unexpectedly, the NADH/NAD$^+$ balance was not affected in the majority of the remaining tested knockout cell lines. This phenomenon may represent metabolic adaptations to mitochondrial dysfunction, such as upregulation of glycolysis, activation of glycerol-phosphate dehydrogenase to by-pass complex I impairment, or overexpression of phosphoenolpyruvate carboxykinase 2 to increase cytosolic pyruvate. These alterations may result in an increased NADH oxidation, with no change of NADH/NAD$^+$ balance (Le et al, 2010; Mráček et al, 2013; Stark and Kibbey, 2014; Stroud et al, 2016). We observed distinct metabolic profiles when comparing NDUFA13-KO and NDUFB4-KO, the cell lines that were the most resistant and most sensitive to cell death, respectively (Fig. EV1E). Taking into consideration this metabolic variety, it is not surprising that complex I impairment did not affect the NADH/NAD$^+$ balance in all knockout cell lines equally. Since the resistance or sensitivity to cell death could be linked to AIFM1 protein levels, we evaluated AIFM1 abundance in the NDUFV3-, NDUFA13-, NDUFA5-, and NDUFB4-KO cells. Surprisingly, the NDUFA13-KO cells overexpressed AIFM1, while no significant change was observed in other knockout cells compared to control HEK293T (Fig. EV1F,G).

Because the NDUFV3- and NDUFA13-KO cell lines displayed the resistance to AIFM1-induced cell death and increased NADH/NAD$^+$ balance, we wondered about the relevance of those complex I accessory subunits for cancer, a pathology marked by cell death resistance. We retrieved the expression profiles of genes that encode mitochondrial complex I accessory subunits using TNMplot from a plethora of cancers (Bartha and Győrffy, 2021) (Fig. 1C, Table EV1). We observed the downregulation of NDUFA13 in a majority of cancers and rather mild effects or even upregulation of NDUFV3 in many cancers. The top two downregulated genes of the complex I accessory subunits were NDUFA7 and NDUFA13. No resistance to AIFM1-induced cell death and no alterations in NADH/NAD$^+$ balance were detected in NDUFA7-KO cells (Fig. EV1A,B,D). On the other hand, NDUFA13-KO cells displayed the resistance to cell death and increased NADH/NAD$^+$ balance together with the observed downregulation in a plethora of cancers (Figs. 1A,B and EV1A,B,D). Therefore, we selected the NDUFA13-KO cells as the model to study the role of MIA40 in the complex with AIFM1.

## AIFM1-MIA40 interaction is increased in NDUFA13-KO cells

We next asked whether the resistance to AIFM1-induced cell death could be related to an increased AIFM1-MIA40 interaction as the NDUFA13-KO cells were marked by an increased NADH/NAD$^+$ balance, which is known to increase the interaction between the two proteins (Hangen et al, 2015). We performed an affinity purification with MIA40$_{FLAG}$ as a bait and confirmed that the AIFM1-MIA40 interaction was increased in NDUFA13-KO

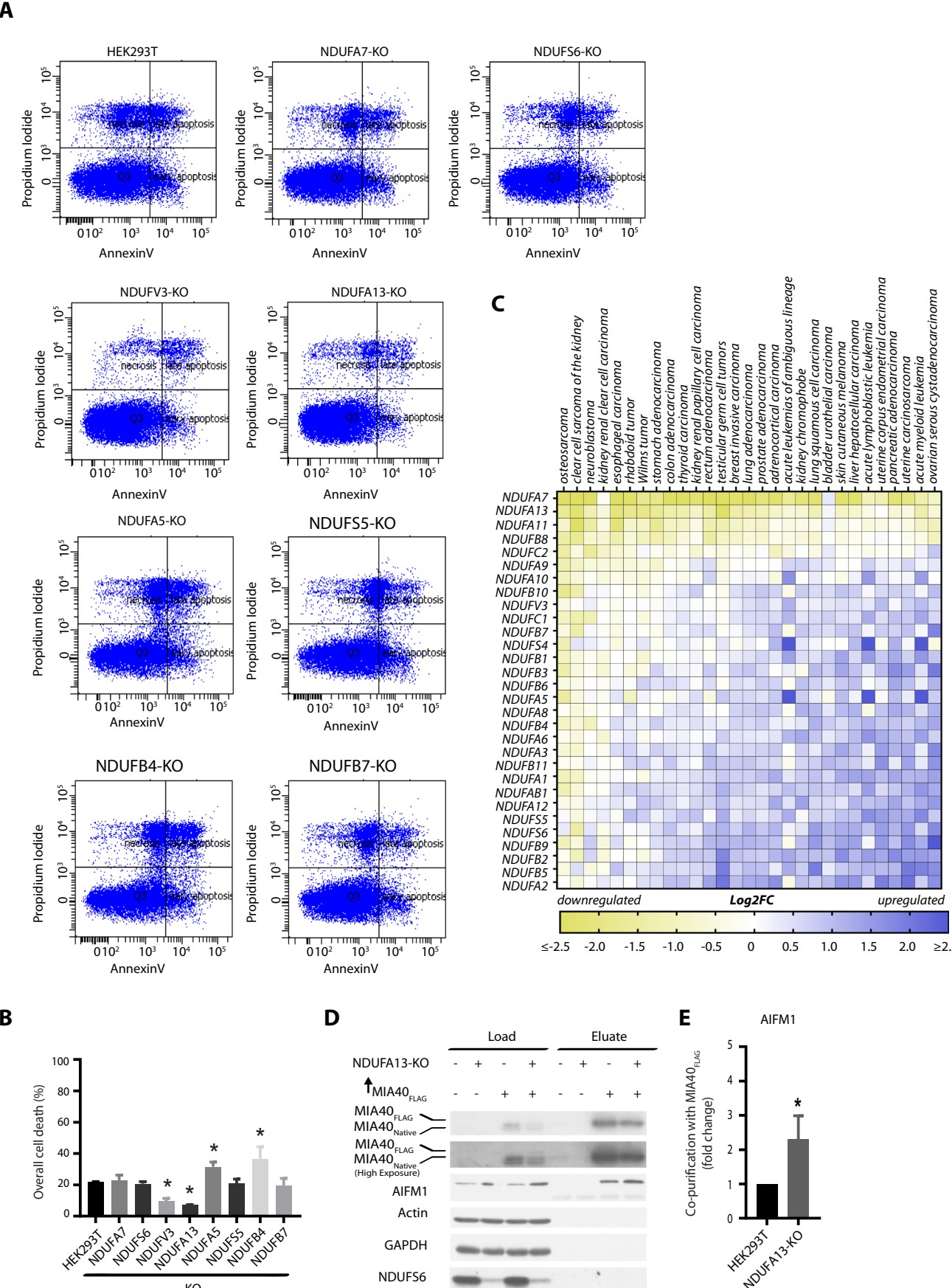

◄ **Figure 1. NDUFA13-KO cells are resistant to AIFM1-induced cell death and have increased AIFM1 and MIA40 interaction.**

(A) Flow cytometer scatter plot analysis of control HEK293T cells and complex I accessory subunit Knock Outs (KO) after the induction of cell death by AIFM1. (B) Quantification of overall cell death of the flow cytometer scatter plot analysis (A). Data shown are mean ± SEM ($n = 3$ biological replicates). Statistical significance was obtained by two-tailed unpaired t-test (compared to HEK293T: NDUFVA7-KO, $p = 0.7833$; NDUFS6-KO, $p = 0.3734$; NDUV3-KO, $p = 0.0015$; NDUFA13-KO, $p < 0.0001$; NDUFA5-KO, $p = 0.0344$; NDUFS5-KO, $p = 0.7440$; NDUFB4-KO, $p = 0.0074$; NDUFB7-KO $p = 0.6396$). *$p < 0.05$ indicates statistical difference from HEK293T. (C) Heatmap of log2-fold changes in median gene expression in cancer vs normal samples. mRNA-seq profiles originated from TCGA, TARGET, GTEx repositories and were retrieved using TNMplot.com tool. Blue and yellow color intensity indicate gene upregulation and downregulation, respectively. (D) Control HEK293T cells or complex I accessory subunit NDUFA13-KO transfected with an empty vector or MIA40$_{FLAG}$ were solubilized, and affinity purification of MIA40$_{FLAG}$ was performed. Fractions were analyzed by SDS–PAGE and Western blot. Load: 3%. Eluate: 100%. (E) Quantification of AIFM1-MIA40 interaction from (D) using ImageJ. Data shown are mean ± SEM ($n = 3$ biological replicates). Statistical significance was obtained by two-tailed unpaired t-test ($p = 0.0286$). *$p < 0.05$ indicates statistical difference from HEK293T. Source data are available online for this figure.

compared to control HEK293T (Fig. 1D,E). Although it was proposed that 90% of MIA40 interacts with AIFM1 in the HEK293 cells (Salscheider et al, 2022), we used the HEK293T cells in our study. These HEK293T cells exhibit a different morphology and doubling time compared to HEK293 cells. Moreover, a substantial body of literature indicates that MIA40 interacts with other clients, without AIFM1 involvement (Habich et al, 2019a; Mohanraj et al, 2019; Petrungaro et al, 2015). In addition, mitochondria isolated from the HEK293T cells, when submitted to swelling, released a significant fraction of free-floating MIA40 to the cytosol (Fig. EV1H). In summary, the NDUFA13-KO cells exhibited the increased NADH/NAD$^+$ balance, resistance to AIFM1-induced cell death, and increased AIFM1-MIA40 interaction. Altogether, this indicates that MIA40 may modulate AIFM1-induced cell death when NADH/NAD$^+$ balance is increased.

Next, we confirmed that NADH led to an increase in the AIFM1-MIA40 interaction by performing an affinity purification using MIA40$_{FLAG}$ as a bait in the absence or presence of 100 μM NADH (Fig. EV2A). We further assessed whether the modulation of NADH/NAD$^+$ balance in the intact cells affects the AIFM1-MIA40 interaction. After the induction of MIA40$_{FLAG}$ expression in the HEK293T cells, the medium was changed to low glucose for 24 h, and then complex I was inhibited with 10 nM or 100 nM rotenone for 12 h. The inhibition of complex I led to an increase in the AIFM1-MIA40 interaction when cells were treated with 10 nM rotenone, but not when cells were treated with 100 nM rotenone (Fig. EV2B). Of note, the higher rotenone concentrations lead to an inhibition of AIFM1 NADH dehydrogenase activity which in turn affect the AIFM1-MIA40 interaction (Elguindy and Nakamaru-Ogiso, 2015). We then confirmed the inhibition of complex I activity and an increase in cellular NADH/NAD$^+$ balance as compared to the control HEK293T cells (Fig. EV2C,D). In summary, we showed that an increased NADH/NAD$^+$ balance promotes the AIFM1-MIA40 interaction.

To clarify whether the resistance to AIFM1-induced cell death observed in the NDUFA13-KO cells was a consequence of NDUFA13 protein loss or of complex I impairment, we rescued the NDUFA13 protein levels and thus complex I activity in the NDUFA13-KO cells. To this end, NDUFA13-KO cells were transfected with a plasmid encoding wild-type NDUFA13 in low glucose for 24, 48, or 72 h and then the cells were shifted to galactose medium for 24 h. Cell survival would indicate the rescue of complex I activity (Fig. EV3A). The rescue of protein levels of NDUFA13 was observed in all groups (Fig. EV3B). However, the NDUFA13-KO cells transfected with an empty vector did not survive the oxidative condition of the galactose medium. Therefore,

additional experiments were performed in the high glucose medium. Next, we checked whether the rescue of NDUFA13 in the NDUFA13-KO cells would alter the cell death resistance phenotype. Indeed, we observed that after 72 h of NDUFA13 rescue, the cells got sensitized to AIFM1-induced cell death (Fig. 2A,B). We also observed a partial complex I reassembly, a partial complex I activity rescue, and finally the restoration of NADH/NAD$^+$ balance after 72 h (Figs. 2C,D and EV3C,E). We then performed the 24 h rescue of NDUFA13 which restored the levels of this protein in NDUFA13-KO cells (Fig. EV3D). However, we did not observe the assembly of complex I, the restoration of NADH/NAD$^+$ balance and the rescue of the cell death phenotype (Figs. 2C, EV3E, 2A). Therefore, the resistance to AIFM1-induced cell death observed in NDUFA13-KO cells is not a consequence of the loss of the NDUFA13 protein, but complex I impairment related to increased NADH/NAD$^+$ balance.

## MIA40 interaction modulates the cell death activity of AIFM1

It was shown before that MIA40 binds the AIFM1 dimer and the MIA40 N-terminal domain is crucial for this interaction (Salscheider et al, 2022). To obtain further structural information, we modeled the MIA40-AIFM1 dimer complex using AlphaFold-Multimer (Evans et al, 2021). The AlphaFold-Multimer models of MIA40 bound to a dimer of AIFM1 demonstrated average protein-level pLDDT scores surpassing 80, thus affirming the accuracy and reliability of these models (Evans et al, 2021). The pTM and ipTM scores were around 0.7 and 0.8, respectively, indicating high accuracy of the predicted structure of the complex (Fig. EV4A–C). To evaluate if the AlphaFold model was able to capture the AIFM1-dimer specific features, we compared the predicted complex structure with experimental structure of the native dimeric form of AIFM1. Superimposition of the predicted structure of MIA40-AIFM1 dimer with the AIFM1 dimer showed agreement in the atomic details (Fig. EV5) (Sevrioukova, 2009). To form the AIFM1(dimer)-MIA40 interaction interface, the N-terminal domain of MIA40 locates underneath AIFM1 C-loop, displacing it and subsequently forming an extended β-sheet with AIFM1 C-terminal domain (Fig. 3A). The formation of this intermolecular beta-sheet was recently shown in the crystal structure of murine AIFM1 in the complex with the N-terminal portion of MIA40 (Fagnani et al, 2024). Remodeling and release of the C-loop were reported to positively correlate with AIFM1 dimerization (Brosey et al, 2016), which is in line with a decreased confidence in C-loop prediction in AIFM1 dimer when compared

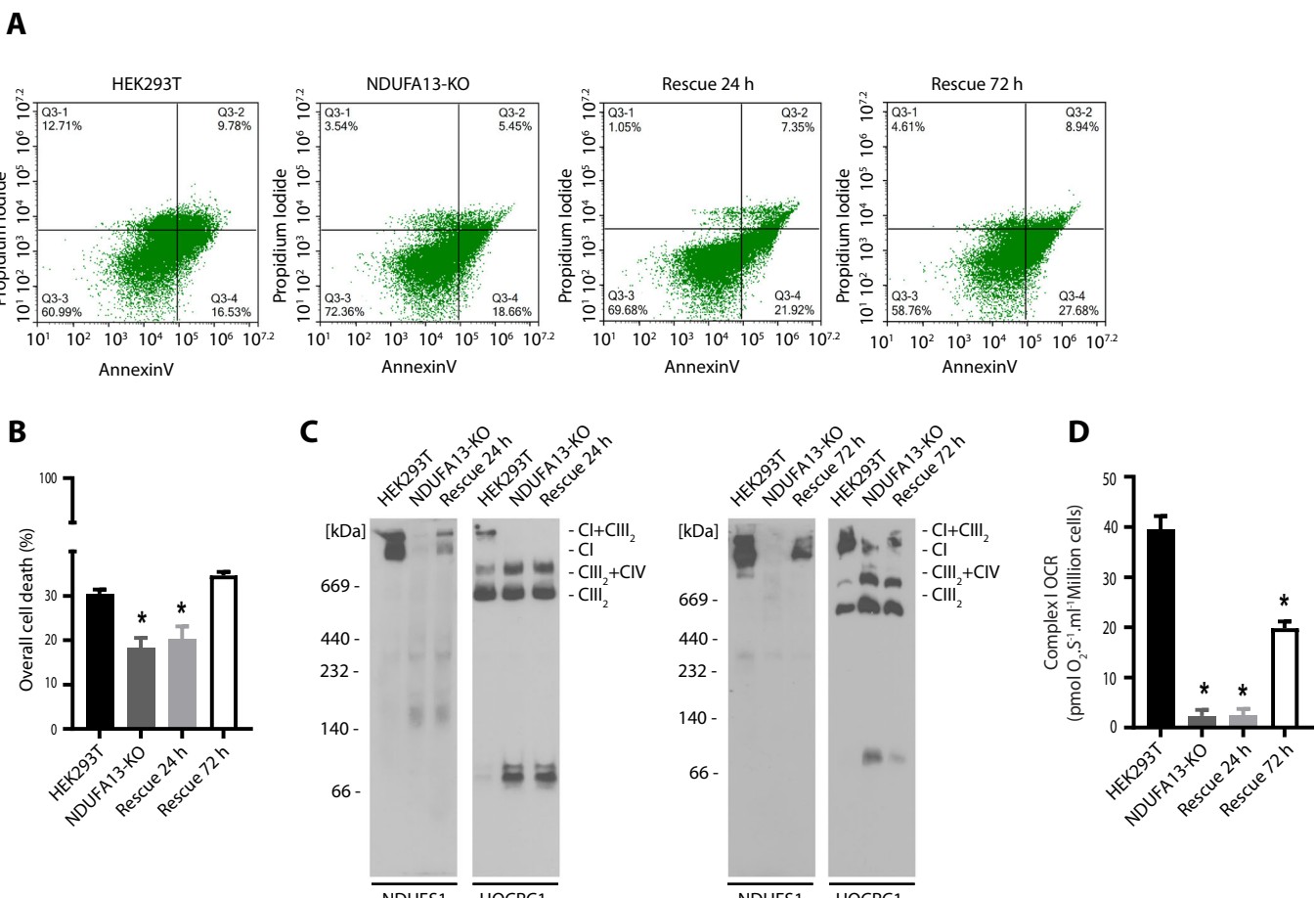

**Figure 2. Rescue of complex I activity sensitizes NDUFA13-KO cells to AIFM1-induced cell death.**

(A) Flow cytometer scatters plot analysis represents the control HEK293T cells, complex I accessory subunit NDUFA13-KO, and NDUFA13 overexpression in NDUFA13-KO after 24 h (Rescue 24 h), or 72 h (Rescue 72 h), in which cell death was induced by AIFM1. (B) Quantification of overall cell death from flow cytometer scatter plot analysis. Data shown are mean ± SEM ($n = 3$ biological replicates). Statistical significance was obtained by ordinary one-way ANOVA with Bonferroni multiple comparisons test (compared to HEK293T: NDUFA13-KO, $p = 0.0055$; Rescue 24 h, $p = 0.0145$; Rescue 72 h, $p = 0.3683$). *$p < 0.05$ indicates statistical difference from HEK293T. (C) Mitochondria isolated from control HEK293T, NDUFA13-KO, Rescue 24 and 72 h, were solubilized in digitonin buffer and analyzed by 3–13% gel BN-PAGE and Western blot. (D) Complex I activity of control HEK293T, NDUFA13-KO, Rescue 24 and 72 h. Data shown are mean ± SEM ($n = 3$ biological replicates). Statistical significance was obtained by one-way ANOVA with Bonferroni multiple comparisons test (compared to HEK293T: NDUFA13-KO, $p < 0.0001$; Rescue 24 h $p < 0.0001$; Rescue 24 h $p = 0.0001$). *$p < 0.05$ indicates statistical difference from HEK293T. Source data are available online for this figure.

to monomeric AIFM1 (Fig. EV4D–F). The prediction of the C-loop region deteriorated even further when MIA40 was present in the structure with AIFM1 (Fig. EV4G). This behavior is in line with recent SAXS and X-ray crystallography structures (Fagnani et al, 2024). Additionally, although MIA40 mostly interacted with one of the monomers of the AIFM1 dimer, the D32 of MIA40 forms hydrogen bonds with R239 of the second monomer of the AIFM1 dimer (Fig. 3B). Our complex prediction showed that MIA40 can contribute to AIFM1 dimer formation and stabilization and is able to conceal the nuclear localization signal (NLS) of AIFM1 (Fig. 3C). Moreover, the predicted MIA40-AIFM1 dimer complex showed a similar conformation of β-hairpin insert (Cβ-clasp) and Trp195/196 as the native AIFM1 dimer with NADH revealed by X-ray crystallography (Fig. 3D,E) (Sevrioukova, 2009). The Cβ-clasp and Trp195/196 are triggering factors for AIFM1 C-loop release and a focal point for AIFM1 dimerization (Romero-Tamayo et al, 2021). Collectively, the AlphaFold prediction shows

that MIA40 can stabilize the AIFM1 dimer and therefore can modulate the translocation of AIFM1 to the nucleus. To further test if MIA40 modulates the AIFM1-induced cell death, we silenced MIA40 in NDUFA13-KO cells (Fig. 3F). The depletion of MIA40 sensitized the NDUFA13-KO cells to AIFM1-induced cell death, indicating that MIA40 reduces the activity of AIFM1 through a physical interaction and stabilization of AIFM1 dimer (Fig. 3G,H).

Next, we investigated whether reducing the AIFM1-MIA40 interaction could sensitize cells to cell death. MIA40 has two cysteine residues (C53 and C55) in its catalytic domain. It has been previously shown that a C55S substitution in MIA40 reduces the interaction of MIA40 with its partners, i.e., Augmenter of Liver Regeneration (ALR), and substrates (Banci et al, 2009; Habich et al, 2019a; Mohanraj et al, 2019). Therefore, we wondered whether the same alteration could reduce the MIA40-AIFM1 interaction and sensitize the cells to the AIFM1-induced

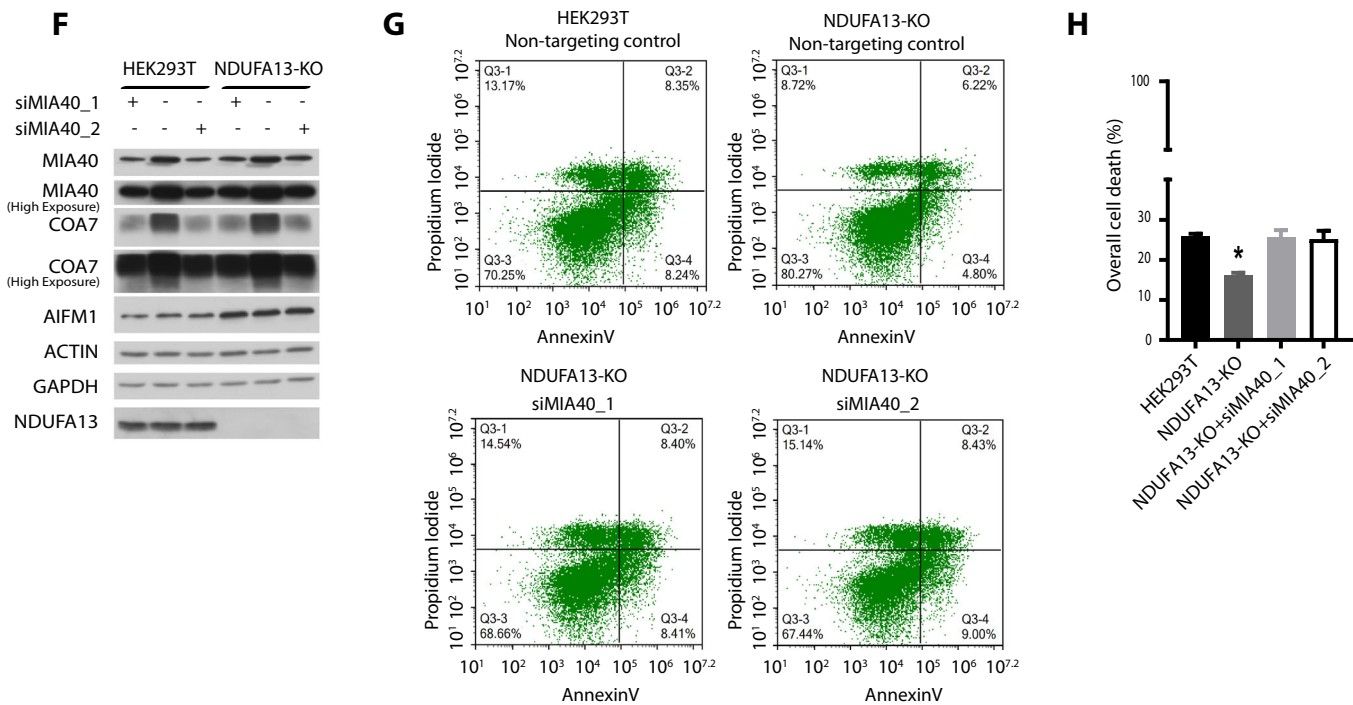

**Figure 3.  MIA40 silencing sensitizes NDUFA13-KO cells to AIFM1-induced cell death.**

(A) Predicted structure of the MIA40-AIFM1 dimer complex. (B) MIA40 binds to one monomer of AIFM1 dimer and forms hydrogen bonds with the second monomer of AIFM1 promoting stabilization of the complex. (C) Nuclear localization signal (NLS, green) of AIFM1 is concealed upon complex formation. (D) In the oxidized monomeric form of mouse AIFM1 (blue), W195 (W196 in human AIFM1) stabilizes the C-terminal loop and inhibits dimerization. (E) In the MIA40-AIFM1 dimer complex (brown and violet), W196 (W195 in mouse AIFM1) changes its conformation similar to reduced NAD-bound AIFM1 (green) which allows dimerization and abolishes AIFM1 cell death activity. (F) Cellular protein extracts were isolated from HEK293T cells, and complex I accessory subunit NDUFA13-KO transfected with two different siRNAs that target different regions of MIA40 mRNA or with siRNA non-targeting control. The samples were analyzed by reducing SDS–PAGE and Western blot. (G) Flow cytometer scatter plot analysis represents the control HEK293T cells and complex I accessory subunit NDUFA13-KO transfected with siRNAs that target different regions of MIA40 mRNA or with siRNA non-targeting control in which cell death was induced by AIFM1. (H) Quantification of overall cell death of flow cytometer scatter plot analysis (F). Data shown are mean ± SEM ($n = 3$ biological replicates). Statistical significance was obtained by an ordinary one-way ANOVA with Bonferroni multiple comparisons test (compared to HEK293T; NDUFA13-KO, $p = 0.0035$; NDUFA13-KO + siMIA40_1, $p = 0.9994$; NDUFA13-KO + siMIA40_2, $p = 0.9735$). *$p < 0.05$ indicates statistical difference from HEK293T cells transfected with siRNA scramble control. Source data are available online for this figure.

cell death. To this end, we induced the expression of MIA40-C55S$_{FLAG}$ in Flp-In T-Rex 293 (MIA40C55S) and observed greater sensitivity to the AIFM1-induced cell death as compared to Flp-In T-Rex 293 expressing MIA40WT$_{FLAG}$ (MIA40WT) (Fig. 4A (the most upper panels) and B). AIFM1-induced cell death is mediated by the RIP1-dependent activation of PARP1 (Jouan-Lanhouet et al, 2012; Xu et al, 2007). To confirm that the cell death observed in MIA40C55S cells is AIFM1-dependent, the cells were treated with either Necrostatin-1 or DPQ, inhibitors of RIP1 and PARP-1 respectively. Both inhibitors were able to reduce the cell death of MIA40C55S cells to approximately 20% (Fig. 4A (middle and last panels) and 4B). Although our results suggest that the majority of the cell death events are related to AIFM1, we cannot exclude that other types of cell death, such as necroptosis, are activated upon the staurosporine treatment. We further confirmed reduced interaction between AIFM1 and MIA40C55S via an affinity purification (Fig. 4C,D). Although the wild-type MIA40 was present in these cells, MIA40C55S had a dominant negative effect over the wild-type protein, possibly by interfering with its normal function and interaction with partners, for example ALR. Of note, the cells expressed relatively equal levels of MIA40WT and MIA40C55S in the total protein fraction and both proteins co-localized with mitochondria (Fig. EV6A,B). To elucidate the structural consequences of the MIA40C55S amino acid substitution and whether it could affect the interaction between AIFM1 and MIA40, we superimposed the predicted structures of wild-type MIA40 and MIA40C55S variant when bound to the AIFM1 dimer. Interestingly, the hydrogen bond between the wild-type MIA40 residue D32 and residue R239 of the second AIFM1 of the dimer was lost upon this alteration (Fig. 4E–G, respectively). Although the use of AlphaFold to predict the effects of single amino acid substitutions and their effects on the protein structure is controversial, our analyses indicated that MIA40C55S could reduce the interaction between MIA40 and the second AIFM1 of the dimer (McBride et al, 2023; Stein and Mchaourab, 2023). Notably, all generated models of the wild-type MIA40 or MIA40C55S with the AIFM1 dimer exhibited ipTM+pTM scores exceeding 1, thus suggesting overall accuracy and reliability (Table EV2). Despite these supportive arguments, only subsequent experimental structural studies on MIA40 and AIFM1 dimer will be able to address this question. In summary, our physiological data together with supportive computational structural data strongly indicate that the weakening of the interaction between MIA40 and AIFM1 sensitizes to the AIFM1-induced cell death.

## AIFM1-induced cell death is modulated by MIA40 in a NADH-dependent manner

Next, we hypothesized that NADH depletion in the IMS might sensitize the NDUFA13-KO cells to AIFM1-induced cell death. To deprive the cells of NADH, we expressed yeast NADH-ubiquinone oxidoreductase 2 (Nde2) in the NDUFA13-KO cells. Yeast have two different external mitochondrial NADH dehydrogenases, Nde1 and Nde2, which catalyze the oxidation of cytosolic NADH. We excluded Nde1 from our experiments because it was reported to trigger cell death on its own by the release of a cytotoxic fragment (Saladi et al, 2020). In yeast, Nde2 is predicted to be anchored to the inner mitochondrial membrane via its N-terminus and to expose a catalytic domain to the IMS. We expressed Nde2-HA in the wild-type HEK293T cells to study its localization in human cells. Nde2-HA was detected in the mitochondrial fraction, and the submitochondrial localization assay confirmed that Nde2 was inserted in the inner mitochondrial membrane with its catalytic domain facing the IMS (Fig. EV7A–C). Subsequently, we expressed Nde2-HA for 48 h in the wild-type HEK293T and NDUFA13-KO cells. In line with our hypothesis, we observed the induction of cell death in NDUFA13-KO cells, whereas the HEK293T cells remained unaffected (Fig. EV7D). Next, we tested whether the modulation of NADH balance would reverse the phenotype of cell death observed in the NDUFA13-KO cells. The Nde2-HA expression was adjusted to result in approximately 50% cell death of NDUFA13-KO cells (Fig. EV7E). Both the HEK293T cells and the remaining living NDUFA13-KO cells expressed similar levels of Nde2-HA after 48 h of transfection (Fig EV7F). Next, we induced cell death via AIFM1 in the remaining living NDUFA13-KO cells after Nde2-HA expression. We observed similar levels of cell death in NDUFA13-KO cells that were transfected with Nde2-HA compared to control HEK293T (Fig. 5A,B). Notably, the expression of Nde2 did not modulate NADH/NAD$^+$ balance under physiological conditions but only when complex I was inhibited (Fig. EV7G), which matches our initial observation of cell death modulation only upon complex I dysfunction.

Finally, we evaluated the effects of Nde2 expression on AIFM1-MIA40 interaction. We transfected Flp-In T-Rex 293 cells with Nde2-HA, and after 24 h we induced MIA40$_{FLAG}$ expression. After 48 h the cells were treated with rotenone. The AIFM1-MIA40 interaction increased upon rotenone treatment in the cells transfected with an empty vector, while upon Nde2-HA expression the AIFM1-MIA40 interaction decreased (Fig. 5C,D). Overall, the expression of Nde2-HA reduced NADH/NAD$^+$ balance and

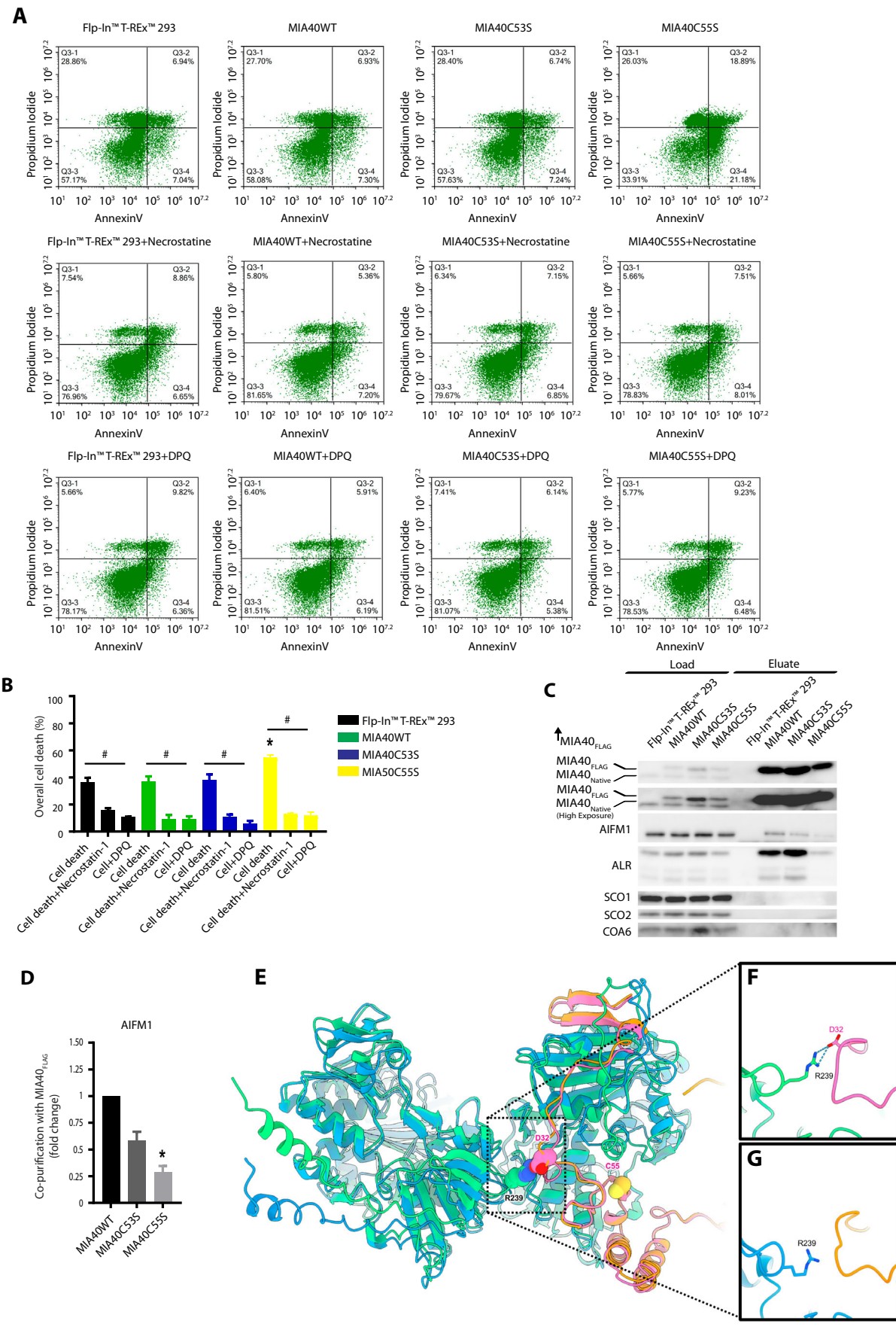

◄ **Figure 4. MIA40C55S substitution reduces AIFM1 and MIA40 interaction and increases sensitivity to cell death.**

(A) Flow cytometer scatter plot analysis represents the Flp-In T-REx293 cells with induced expression of wild-type or variants of MIA40$_{FLAG}$ C53S or C55S in which cell death was induced by AIFM1 and subsequently these cells were treated with a vehicle, Necrostatine-1 or DPQ. (B) Quantification of overall cell death of flow cytometer scatter plot analysis (A). Data shown are mean ± SEM ($n = 3$ biological replicates) * and # $p < 0.05$ from an ordinary two-way ANOVA with Bonferroni multiple comparisons test. * Difference compared to Flp-In T-REx293 cells after induced cell death by AIFM1 (MIA40 WT, $p > 0.9999$; MIA40C53S, $p > 0.9999$; MIA40C55S, $p = 0.0020$). # Difference among each cell line treated with vehicle, Necrostatine-1 or DPQ after induced cell death by AIFM1 (For Flp-In T-REx293; Necrostatine-1, $p = 0.0004$; DPQ, $p < 0.0001$. For MIA40 WT: Necrostatine-1, $p < 0.0001$; DPQ, $p < 0.0001$. For MIA40C53S: Necrostatine-1, $p < 0.0001$; DPQ, $p < 0.0001$. For MIA40C55S: $p < 0.0001$; DPQ, $p < 0.0001$). (C) Cellular protein extracts from Flp-In T-REx293 cells expressing wild-type (MIA40WT), or variants of MIA40$_{FLAG}$ C53S or C55S (MIA40C53S or MIA40C55S), were subjected to an affinity purification via MIA40$_{FLAG}$ as a bait. Load and eluate fractions were analyzed by reducing SDS–PAGE and Western blot. Load: 2.5%. Eluate: 100%. (D) Quantification of AIFM1-MIA40 interaction from (C) using ImageJ. Data shown are mean ± SEM ($n = 3$ biological replicates) Statistical significance was obtained by an ordinary one-way ANOVA with Bonferroni multiple comparisons test (compared to MIA40WT: MIA40C53S $p = 0.0702$; MIA40C55S $p = 0.0075$). *$p < 0.05$ indicates statistical difference from Flp-In T-REx293 cells with induced expression of MIA40 wild-type. (E) Superimposed predicted structures of wild-type MIA40 (pink) and MIA40C55S variant (orange) bound to AIFM1 dimer (green and blue, respectively). The side chains of MIA40 D32 forming the hydrogen with R239 in the second AIFM1 monomer in the AIFM1 dimer are shown as ball representations. (F, G) The hydrogen bond formed between MIA40 and the second monomer in the AIFM1 dimer is lost upon MIA40C55S substitution. Source data are available online for this figure.

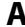

**Figure 5. Expression of yeast NADH-ubiquinone oxidoreductase 2 sensitizes NDUFA13-KO cells to AIFM1-induced cell death.**

(A) Flow cytometer scatter plot analysis represents the control HEK293T cells and complex I accessory subunit NDUFA13-KO transfected with an empty vector or Nde2-HA in which AIFM1 cell death was induced. (B) Quantification of overall cell death of flow cytometer scatters plot analysis (A). Data shown are mean ± SEM ($n = 3$ biological replicates) Statistical significance was obtained by an ordinary one-way ANOVA with Bonferroni multiple comparisons test (compared to HEK293T: HEK293T + Nde2-HA; $p = 0.9994$; NDUFA13-KO; $p = 0.0172$; NDUFA13-KO + Nde2-HA $p = 0.9882$). *$p < 0.05$ indicates statistical difference from HEK293T. (C) Affinity purification of MIA40$_{FLAG}$ from cellular lysates extracted from Flp-In T-REx293 cells in which the expression of MIA40$_{FLAG}$ was induced. Cells were transfected with an empty vector or Nde2-HA 24 h prior to MIA40$_{FLAG}$ induction. After 24 h of MIA40$_{FLAG}$ induction, cells were treated with a vehicle or rotenone during 12 h. (ˆ) The antibody against HA decorates an artifact in the Flp-In T-REx293 cells transfected with an empty vector in load fractions which is at the same size as Nde2-HA. (#) HA antibody artifact. Load and eluate fractions were analyzed by reducing SDS–PAGE and Western blot. Load: 4%. Eluate: 100%. were subjected to affinity purification. Load and eluate fractions were analyzed by reducing SDS–PAGE and Western blot. Load: 2.5%. Eluate: 100%. (D) Quantification of AIFM1-MIA40 interaction from (C) using ImageJ. Data shown are mean ± SEM ($n = 3$ biological replicates). Statistical significance was obtained by two-tailed unpaired t-test. For MIA40 signal compared to Flp-In T-REx293: Rotenone, $p = 0.0471$; Rotenone + Nde2-HA, $p = 0.0007$. For COA7 signal compared to HEK293T: Rotenone $p = 0.0471$; Rotenone + Nde2-HA, $p = 0.0655$. *$p < 0.05$ indicates statistical difference from Flp-In T-REx293 cells with induced expression of MIA40$_{FLAG}$ treated with a vehicle. Source data are available online for this figure.

subsequently the AIFM1-MIA40 interaction, which sensitized the NDUFA13-KO cells to the AIFM1-induced cell death. Altogether, these findings support our finding that the interaction of MIA40 and AIFM1 modulates AIFM1-induced cell death in an NADH-dependent manner.

## Discussion

We propose that MIA40 is a modulator of AIFM1-induced cell death under metabolic alterations resulting in increased NADH/NAD$^+$ balance (Bano and Prehn, 2018; Herrmann and Riemer, 2021). We observed this phenomenon in a model of complex I impairment, the NDUFA13-KO cells, which are characterized by high NADH levels and an increased AIFM1-MIA40 interaction. We showed that the depletion of NADH levels reduced the interaction between MIA40 and AIFM1, which in turn sensitized NDUFA13-KO cells to AIFM1-induced cell death. Our AlphaFold model predicted contribution of MIA40 to AIFM1 dimerization and stabilization confirming previous observations (Salscheider et al, 2022). Moreover, the AIFM1-MIA40 interaction conceals the NLS of AIFM1, potentially inhibiting AIFM1 nuclear translocation (Sevrioukova, 2011; Churbanova and Sevrioukova, 2008). Therefore, the interaction of MIA40 with AIFM1 may function to inhibit its cell death activity. Accordingly, the impairment of MIA40 sensitized NDUFA13-KO cells to AIFM1-induced cell death.

MIA40 is crucial for importing subunits of complex I, such as NDUFS5, NDUFB7, and NDUFA8 (Angerer et al, 2011; Mimaki et al, 2012). The NADH/NAD$^+$ balance is a metabolic sensor related to complex I impairment (Titov et al, 2016). Therefore, an increase in the NADH/NAD$^+$ balance could in turn enhance AIFM1-MIA40 interaction, possibly to improve MIA40 pathway efficiency to recover complex I biogenesis and the reorganization of cellular metabolism as recently proposed (Herrmann and Riemer,

2021; Salscheider et al, 2022). Perturbations of complex I cause metabolic stress, but not all instances of metabolic stress are detrimental. To allow the cell to regain homeostasis, an interaction between MIA40 and AIFM1 may function to prevent premature cell death upon complex I impairment and increased NADH/NAD$^+$ balance as observed in the NDUFA13-KO cellular model (Perier et al, 2005).

The role of complex I in cancer is controversial (Urra et al, 2017). Complex I subunits may act as cancer suppressors (Iommarini et al, 2014), and mutations of complex I genes result in the progression of tumorigenesis (Gorelick et al, 2021). Our proposed role of MIA40 modulation of AIFM1-induced cell death would be most suited to be tested on cancers that exhibit high NADH/NAD$^+$ balance levels such as breast and colon cancers (Santidrian et al, 2013). NDUFA13 is a master regulator of metastasis, tumor suppression, and metabolism (Nallar and Kalvakolanu, 2017). So far, NDUFA13 has been found to modulate cell death induced via caspase activation (Fearnley et al, 2001; Ma et al, 2007; Verhagen et al, 2002). Here, we show for the first time that cells lacking NDUFA13 are more resistant to cell death independent of caspase activation.

Our findings may help understand survival strategies of cancers in which NDUFA13 is downregulated, including, but not limited to kidney, prostate, gastrointestinal, lung, cervix, ovary, liver, nervous system, adrenal, and breast cancer (Nallar and Kalvakolanu, 2017). Our results suggest a new survival strategy for stress conditions related to elevated NADH/NAD$^+$ balance, namely, the escape from cell death through MIA40 physical modulation of AIFM1 activity. Future studies may apply our findings to design cancer therapies that target the AIFM1-MIA40 physical interaction in the presence of elevated NADH/NAD$^+$ balance.

## Methods

**Reagents and tools table**

| Reagent/Resource | Reference or Source | Identifier or Catalog Number |
| --- | --- | --- |
| **Experimental models** | | |
| HEK293T Wild Type | Stroud et al, 2016 | N/A |
| HEK293T NDUFA7-KO | Stroud et al, 2016 | N/A |
| HEK293T NDUFS6-KO | Stroud et al, 2016 | N/A |

| Reagent/Resource | Reference or Source | Identifier or Catalog Number |
|---|---|---|
| HEK293T NDUFV3-KO | Stroud et al, 2016 | N/A |
| HEK293T NDUFA13-KO | Stroud et al, 2016 | N/A |
| HEK293T NDUFA5-KO | Stroud et al, 2016 | N/A |
| HEK293T NDUFS5-KO | Stroud et al, 2016 | N/A |
| HEK293T NDUFB4-KO | Stroud et al, 2016 | N/A |
| HEK293T NDUFB7-KO | Stroud et al, 2016 | N/A |
| Flp-In T-REx 293 | Mohanraj et al, 2019 | N/A |
| Flp-In T-REx 293 MIA40flag Wild Type | Mohanraj et al, 2019 | N/A |
| Flp-In T-REx 293 MIA40flag C53S | Mohanraj et al, 2019 | N/A |
| Flp-In T-REx 293 MIA40flag C55S | Mohanraj et al, 2019 | N/A |
| **Recombinant DNA** | | |
| **Antibodies** | | |
| Actin | Sigma-Aldrich | A1978: RRID: AB_476692 |
| AIF | Abcam | Ab1998: RRID:AB_302748 |
| ALR | Santa Cruz Biotechnology | Sc-134869; RRID:AB_10608344 |
| ATP5A | Abcam | Ab14748; RRID:AB_301447 |
| CHCHD4 | Protein | 21090-1-AP; RRID: AB_10734583 |
| COX6A | Rabbit serum | In-house produced, a gift from Prof. P. Rehling |
| COA7 | Sigma-Aldrich | HAP029926; RRID:AB_10602173 |
| GAPDH | Santa Cruz Biotechnology | SC-47724: RRID:AB_627678 |
| Anti-HA | Sigma-Aldrich | H9658: RRID:AB_260092 |
| MIA40 | Rabbit Serum | WA136-5 |
| Monoclonal ANTI-FLAG M2 | Sigma-Aldrich | F1804; RRID: AB_262044 |
| NDUFA13 | Thermo Fisher | 6E1BH7; RRID: AB_2533544 |
| NDUFS6 | Abcam | Ab195808 |
| NDUFS1 | Santa Cruz Biotechnology | Sc50132; RRID:AB_2298398 |
| SCO1 | Rabbit Serum | In-house produced, a gift from Prof. P. Rehling |
| SCO2 | Rabbit Serum | In-house produced, a gift from Prof. P. Rehling |
| TOMM40 | Santa Cruz Biotechnology | Sc-365467; RRID:AB_10847086 |
| TOMM20 | Santa Cruz Biotechnology | Sc-11415; RRID:AB_2207533 |
| TOMM22 | Abcam | Ab-179826; RRID: AB_3095411 |
| CHCHD2 | Proteintech | 19424-1-AP; RRID: AB_10638907 |
| SDHA | Santa Cruz Biotechnology | Sc-166947; RRID:AB_10610526 |
| Tubulin | Santa Cruz Biotechnology | Sc-134239; RRID:AB_2210215 |
| UQCRC1 | Sigma-Aldrich | HPA002815; RRID:AB_1080486 |
| Aconitase 2 | Cell Signaling Technology | 6571; RRID:AB_2797630 |
| TIMM17B | Thermo Fisher Scientific | PA5-56054; RRID:AB_2648475 |
| **Oligonucleotides and other sequence-based reagents** | | |
| Nde2-HA forward primer 5'-aggccggatccATGCTGCCAAGACTCGGC-3 | This study | N/A |
| Nde2-HA reverse primer 5-ggcctctcgagTCAAGCGTAATCTGGAACATCGTATGGGTAGACACTGGAATCCCTGCCC-3' | This study | N/A |

| Reagent/Resource | Reference or Source | Identifier or Catalog Number |
|---|---|---|
| NDUFA13 forward primer<br>5'-GAATTC ATG GCG GCG TCA AAG GTG AAG CA-3' | Sigma | PDSIRNA2D (siRNA ID: SASI_Hs01_00192719) |
| NDUFA13 reverse primer<br>5'-TTTT TCTAGA TCA CGT GTA CCA CAT GAA GCC GT-3' | Sigma | PDSIRNA2D (siRNA ID: SASI_Hs01_00192719) |
| MIA40_siRNA_1<br>5'-ATAGCACGGAGGAGATCAA-3 | Mohanraj et al, 2019 | N/A |
| MIA40_siRNA_2<br>5'-GGAATGCATGCAGAAATAC-3' | Mohanraj et al, 2019 | N/A |
| **Chemicals, Enzymes and other reagents** | | |
| Tetracycline | Roch | 64-75-5 |
| Gene Juice Transfection Reagent | MerckMillipore | 70967 |
| Oligofectamine | MerckMillipore | 12252011 |
| Digitonin, High Purity | Calbiochem | 300410 |
| MitoPlate S-1 Biolog Phenotype MicroArray MicroPlates | Biolog Inc. | MS 14105 |
| Biolog Redox Dye Mix MC (6x) | Biolog Inc. | 74353 |
| Biolog MAS (2x) | Biolog Inc. | 72303 |
| Staurosporine, Streptomyces sp. | Merck | 569396 |
| necrostatin-1 | Merck | 480065 |
| DPQ, PARP-1 inhibitor | Abcam | ab141461 |
| Z.VAD.FMK | Sigma | V116 |
| NAD/NADH Assay Kit (Fluorometric) | Abcam | ab176723 |
| Dead Cell Apoptosis Kits with Annexin V for Flow Cytometry | Invitrogen™ | V13245 |
| Rotenone | Sigma | R8875 |
| ANTI-FLAG® M2 Affinity Gel | Sigma | A2220 |
| CellTiter 96® AqueousOne Solution Cell Proliferation (MTS) | Promega | G3582 |
| Sulforhodamine B | Sigma | S9012 |
| **Software** | | |
| Graphpad prism | Dodtmacs | N/A |
| DatLab v. 5.1.0.20 | Oroboros Instruments, Innsbruck, Austria | N/A |
| FACSDiva software | BD bioscience | N/A |
| Novo Express | Agilient | N/A |
| FlowJow | BD bioscience | N/A |
| Zeiss Zen 3.7 | Zeiss | |
| ImageJ | Open source | N/A |
| **Instrument** | | |
| Oxygraph-2K | Oroboros Instruments | |
| BD LSR Fortessa Cell Analyser equipped with the 355 nm, 405 nm, 488 nm, and 635 nm laser | BD bioscience | |
| NovoCyte flow cytometer | Agilient | |
| Axio observer inverted microscope | Zeiss | |

## Cell lines and growth condition

We used the following cells kindly gifted by Dr. Michael T. Ryan (Stroud et al, 2016) from Monash Biomedicine Discovery Institute, Monash University, 3800, Melbourne, Australia: HEK293T WT and the complex I-deficient cell lines NDUFA7-KO, NDUFS6-KO, NDUFV3-KO, NDUFA13-KO, NDUFA5-KO, NDUFS5-KO, NDUFB4-KO and NDUFB7-KO. Cells were cultured in Dulbecco's modified Eagle's medium (DMEM) with high-glucose content (4500 mg/L) supplemented with 10% (v/v) fetal bovine serum (FBS), 2 mM L-glutamine, 1% (v/v) penicillin-streptomycin and 50 µg/ml uridine at 37 °C in a 5% $CO_2$ incubator. We also used Flp-In T-REx 293 cells with inducible expression of MIA40$_{FLAG}$, MIA40-C53S$_{FLAG}$ or MIA40-C55S$_{FLAG}$, which were cultured in the

same conditions as specified above excluding 50 µg/ml uridine. Expression of respective proteins was induced by treatment with tetracycline (100 ng/ml) for 24 h (Mohanraj et al, 2019). Where indicated, cells were cultured in DMEM that contained low glucose (1.1 g/l) or galactose (1.8 g/l).

## Antibodies

The antibodies against the following proteins were used in the study: actin (Sigma, A1978, 1:500), AIF (Abcam, ab1998, 1:500), ALR (Santa Cruz Biotechnology, Sc-134869, 1:500), ATP5A (Abcam, ab14748, 1:500), COX6A (Rabbit Serum, 3282.7, 1:1000), COA7 (Sigma, HPA029926, 1:500), GAPDH (Santa Cruz Biotechnology, Sc-47724, 1:1000), HA, MIA40 (Rabbit Serum, WA136-5,1:500), NDUFA13 (Thermo Fisher, 6E1BH7, 1:500), NDUFS6, NDUFS1 (Santa Cruz Biotechnology, Sc-50132 1:1000), SCO1, SCO2, TOMM40 (Santa Cruz Biotechnology, Sc-365467, 1:500), TOMM20 (Santa Cruz Biotechnology, Sc-11415, 1:500), TOM22 (Abcam, ab179826, 1:500), CHCHD2 (Proteintech, 19424-1-AP 1:500), SDHA (Santa Cruz Biotechnology, Sc-166947, 1:1000), Tubulin, UQCRC1 (Sigma, HPA002815, 1:500), Aconitase 2 (Cell Signaling, 6571, 1:500), TIMM17B (Invitrogen, PA5-56054, 1:500).

## DNA and siRNA transfection

Protein overexpression and silencing was performed as described previously (Mohanraj et al, 2019), using Gene Juice Transfection Reagent (70967; MerckMillipore) or oligofectamine (12252011; Merck-Millipore) according to the manufacturer's protocol. DNA sequence encoding yeast gene NDE2 was taken from The Saccharomyces Genome Database (SGD). The DNA sequence of Nde2 was codon optimized for *H. Sapiens* and cloned with C-terminally localized HA tag (TAC CCA TAC GAT GTT CCA GAT TAC GCT) in pcDNA 3.1 (+) between BamHI and XhoI restriction sites. NDUFA13WT sequence was obtained from Sigma. To generate an expression vector for NDUFA13, cDNA fragment containing the entire protein encoding ORF of NDUFA13 was amplified with a forward primer containing a EcoRI site (5'-GAATTC ATG GCG GCG TCA AAG GTG AAG CA-3', Sigma) and a reverse primer containing sequences specifying a stop codon, and an XbaI site (5'-TTTT TCTAGA TCA CGT GTA CCA CAT GAA GCC GT-3', Sigma). The PCR products were digested with EcoRI and Xba1 restriction enzymes and cloned into the EcoRI and Xba1-digested pcDNA3.1/Zeo(+) vector. Cloned construct was sequenced for insert verification. For MIA40 gene knockdown the following siRNA sequences were used: MIA40_1(5'-ATAGCACGGAGGAGATCAA-3') or MIA40_2(5'-GGAATGCATGCAGAAATAC-3') in accordance to our previous publication. Control cells were transfected with Mission siRNA universal negative control (SIC001, Sigma).

## Total cellular protein extract

Radioimmunoprecipitation assay (RIPA) was used as described previously (Mohanraj et al, 2019). The harvested cells were incubated in RIPA buffer (65 mM Tris–HCl [pH 7.4], 150 mM NaCl, 1% v/v NP40, 0.25% sodium deoxycholate, 1 mM ethylene-diaminetetraaceticacid [EDTA], and 2 mM phenylmethylsulfonyl fluoride [PMSF]) for 30 min at 4 °C. Clarification of the lysate was performed by centrifugation at $14,000 \times g$ for 30 min at 4 °C. The supernatant was collected and the protein concentration was measured. The 2×Laemmli buffer with 50 mM DTT was used to dilute the supernatant.

## Mitochondria Isolation and subcellular fractionation

Harvested cells were subjected to centrifugation at $1000 \times g$ for 5 min at 4 °C after resuspension in isotonic buffer 4 °C (10 mM MOPS [pH 7.2], 75 mM mannitol, 225 mM sucrose, and 1 mM EGTA) supplemented with 2 mg/ml BSA and 2 mM PMSF. The pellet was resuspended and incubated for 5 to 7 min at 4 °C in hypotonic buffer (10 mM MOPS [pH 7.2], 100 mM sucrose, and 1 mM EGTA) prior to homogenization using a Dounce glass homogenizer (Sartorius). Osmolarity was re-established using hypertonic buffer (1.25 M sucrose and 10 mM MOPS [pH 7.2] - 1.1 ml/g of cells). To remove cellular debris, the homogenate was centrifuged at $1000 \times g$ for 10 min at 4 °C. The supernatant containing mitochondria was subjected to further centrifugation at $10,000 \times g$ for 10 min at 4 °C. The mitochondrial pellet was resuspended in isotonic buffer without BSA and proteins were quantified using the Bradford assay. Mitochondria were resuspended in 2×Laemmli buffer with 50 mM DTT.

Subcellular fractionation was performed as follows: The post-nuclear supernatant was divided into three equal aliquots after homogenization. One aliquot was labeled as the total, the other two were centrifuged at $10,000 \times g$ for 10 min at 4 °C. The supernatant of one sample was labeled as cytosol and the pellet of the other as mitochondria. The pyrogallol red was used to precipitate proteins from the total and cytosolic fractions, all three fractions were denatured in urea sample buffer with 50 mM DTT for SDS–PAGE and Western blot.

## Blue-Native PAGE

Mitochondria were solubilized in digitonin buffer (1% [wt/vol] digitonin, 20 mM Tris–HCl, pH 7.4, 50 mM NaCl, 10% [wt/vol] glycerol, 0.1 mM EDTA, 1 mM PMSF) at concentration of 1 µg of mitochondria/1 µl of digitonin buffer for 15 min at 4 °C. Then, the lysate was centrifuged at $20,000 \times g$ for 15 min at 4 °C. The samples were resolved at 4 °C using a gradient gel (4–13%). The High Molecular Weight Calibration Kit for native electrophoresis (Amersham) was used as a molecular weight standard.

## Mitoplast

Mitochondria were incubated on ice for 30 min in swelling buffer (25 mM sucrose and 20 mM HEPES/KOH [pH 7.4]), or sucrose buffer (250 mM sucrose and 20 mM HEPES/KOH [pH 7.4]) and divided in two equal parts. The first part was treated with proteinase K (25 µg/ml) for 5 min on ice. The second part was untreated. Subsequently, Proteinase K was inactivated by 2 mM PMSF. Mitochondria were pelleted by centrifugation at $10,000 \times g$ for 10 min at 4 °C. The pellet was diluted in 2xLaemmli buffer containing 50 mM DTT and analyzed by SDS–PAGE and subsequent Western blot.

## MitoPlates S-1

Mitochondrial function assays using MitoPlates S-1 were performed as suggested by the manufacturer. Briefly, assay mix was

dispensed into all wells of a MitoPlate and then the plate was incubated at 37 °C for 1 h to allow substrates to fully dissolve. Control HEK293T, NDUFA13-KO and NDUFB4-KO were harvested ($1.8 \times 10^6$ cells of each cell line), pelleted and resuspended in 1× Biolog Mitochondrial Assay Solution (MAS) containing saponin. Then, $45 \times 10^3$ cells were dispensed into each well of a MitoPlate (final concentration of saponin 30 μg/ml). Color formation at 590 nm was read kinetically for 4 h on a Multiskan FC Microplate Photometer (Thermo Scientific). The background was corrected for the blank sample and the average rate between 1 h and 3 h was calculated. The protein content was measured using Bradford method. Results are presented as average rate per minute per μg of protein.

## FLAG-tag affinity purification

Cells were harvested and pelleted by centrifugation and subsequently resuspended in PBS supplemented with 2 mM PMSF and 50 mM IAA. The cells were then solubilized in lysis buffer (50 mM Tris–HCl [pH 7.4], 150 mM NaCl, 10% glycerol, 1 mM EDTA, 2 mM PMSF, 50 mM IAA and 1% digitonin) for 20 min at 4 °C. The lysate was clarified by centrifugation at $20,000 \times g$ for 15 min at 4 °C. After load aliquot preparation (2 to 5%), the supernatant was incubated with anti-FLAG M2 affinity gel (Sigma) for 2 h at 4 °C under mild rotation. After binding, the samples were centrifuged at $200 \times g$ for 5 min at 4 °C, unbound aliquot was prepared (2 to 5%), and the resin was washed five times with lysis buffer without digitonin. The resin-bound proteins were eluted with Laemmli buffer containing 50 mM DTT, incubated at 65 °C for 4 min and subjected to centrifugation at $6000 \times g$ for 1 min at RT. Of note, non-transfected Flp-In T-Rex 293 cells exhibit a faint artifact signal in the same molecular weight as the cells transfected with Nde2-HA in the load fraction, whereas HEK293T cells do not exhibit the same issue (Fig. 5C and Expanded View Fig. 7f).

## Complex I activity

Oxygraph 2k high resolution respirometer (Oroboros Instruments, Innsbruck, Austria) was used to access complex I activity by substrate-driven respiration protocol in digitonin-permeabilized cells as previously described (Chojnacka et al, 2022). The CI activity was stimulated by malate (0.5 mM), pyruvate (5 mM), and glutamate (10 mM) in the presence of $ADP^+Mg^{+2}$ (2.5 mM) with subsequent inhibition by rotenone (0.05 μM). The CIII activity was stimulated by succinate (10 mM) and glycerol-3-phosphate (5 mM) and outer mitochondrial membrane integrity was accessed by cytochrome c (10 μM) with subsequent inhibition of CIII by antimycin A (2.5 μM). The CIV activity was stimulated by ascorbate (2 mM) and N,N,N′, N′-tetramethyl-p-phenylenediamine dihydrochloride (TMPD; 0.5 mM) with subsequent inhibition by sodium azide (50 mM). DatLab v. 5.1.0.20 (Oroboros Instruments, Innsbruck, Austria) was used to record and analyze the data. During the measurement, the cells were resuspended in MiR05 medium (ethylene glycol-bis (β-aminoethyl ether)-N,N,N′,N′tetraacetic acid [EGTA] 0.5 mM, MgCl₂ 3 mM, lactobionic acid 60 mM, taurine 20 mM, KH₂PO₄ 10 mM, HEPES 20 mM, d-sucrose 110 mM, 1 mg/ml BSA–fatty acid free added freshly, pH 7.1).

## NADH/NAD⁺ balance

Total NADH/NAD⁺ levels were measured accordingly to manufacturer instruction (AB176723, NAD⁺/NADH Assay Kit, Abcam) and as previously described in the literature (Yeo et al, 2020). Detection was obtained at 567 nm.

## Microscopy

Contrast microscopy images of living cells plated in 6-well plates were obtained with Axio observer inverted microscope Zeiss. Data was collected using Zeiss Zen 3.7 software. Images were obtained by transmitted light at 20× magnification and the intensity was automatically adjusted by the software.

## Cell death

For viability and survival experiments, $2 \times 10^4$ cells of each cell line were plated in 96 well plates. On the following day, medium was changed to avoid false positive cell death events, and cells were pre-treated with 100 μM of z-vad-fmk, the pan-caspase inhibitor, for 30 min. Next, cells were stimulated with 2 μM staurosporine for 3.5 h to induce cell death. The z-vad-fmk 100 μM was kept in the medium during the total 4 h treatment (Daugas et al, 2000). Cell viability was accessed by CellTiter 96® AqueousOne Solution Cell Proliferation Assay (Promega Corporation), following manufactory instructions, reaction time lasted 1 h. Sulforhodamine B colorimetric assay for cytotoxicity was used to access cell survival accordingly to previous publication (Vichai and Kirtikara, 2006). Cell death was assessed by annexinV (AV) and propidium iodide (PI) staining using Alexa Flour™ 488 AnnexinV/Dead cell apoptosis kit (Thermo Fisher Scientific) following manufacturer instructions. Data was acquired using a BD LSR Fortessa or Agilent NovoCyte flow cytometers to measure percentages of Annexin V and PI positive cells to compare cell death among different HEK293T and the other complex I accessory subunit KOs used in this study (Fig. 1A). Data was analyzed using FlowJo or Novo Express software as described previously (Cieśla et al, 2021). For the confirmation of cell death phenotype related to AIFM1, cells were pre-treated with 60 μM Necrostatin-1 for 1 h or 30 μM DPQ for 14 h. All data was normalized to the respective vehicle control (DMSO 0.2%).

## Gene expression profiling

Expression profiles of genes encoding for mitochondrial complex I core accessory subunits were retrieved using TNMplot.com (Bartha and Győrffy, 2021). The retrieved RNA-seq data originated from The Cancer Genome Atlas (TCGA), Therapeutically Applicable Research to Generate Effective Treatments (TARGET), and The Genotype-Tissue Expression (GTEx) repositories. Heat maps were generated using Prism v8.4.3 (GraphPad Software, USA) running on a personal computer.

## Modeling of dimer AIFM1 in complex with MIA40

MIA40 (UniProt ID-Q8N4Q1) and AIFM1 (UniProt ID-O95831) protein sequence were downloaded from the UniProt sequence database (https://www.uniprot.org/). The truncated form of AIFM1

1-102Δ was used to avoid interaction between AIFM1 transmembrane domain and MIA40 hydrophobic cleft. AlphaFold-multimers (Evans et al, 2021; Mirdita et al, 2022), was used to predict the protein complex structures and a truncated version of AIFM1 was used to avoid interaction between MIA40 hydrophobic cleft and AIFM1 transmembrane domain. AlphaFold-Multimer, uses a model trained on the protein structure database and multiple sequence alignments to infer the structure of proteins and multiprotein complex. The quality of the predicted atomic models was evaluated based on AlphaFold prediction quality metrices: predicted local-distance difference test (pLDDT), predicted template modeling (pTM) score, interface predicted template modeling (ipTM) score, and Predicted Alignment Error (PAE). The pLDDT provides a confidence measure for the per-residue accuracy of the predicted structure, the pTM assesses overall structure accuracy, while ipTM evaluate accuracy of inter-subunit interaction interfaces, and PAE provides measure for the confidence in the relative positions and orientations of parts of the predicted structure. Nineteen models of the MIA40-AIFM1 complex were generated, with detailed pTM, ipTM, and pLDDT scores available in Table EV2. UCSF ChimeraX, was used to visualize and analyze the predicted structure of the dimer AIFM1-Mia40 complex (Goddard et al, 2018; Pettersen et al, 2021; Pettersen et al, 2004). The quality of AlphaFold predicted, confidence and analysis were performed using ChimeraX and PDBePISA (Krissinel and Henrick, 2007).

## Statistics

All numerical data are expressed as a mean ± SEM (standard error) and analyzed by the one-way or two-way ANOVA followed by the Bonferroni's test as post hoc for multiple group analysis. Two-tailed unpaired t-test was used for accessing difference between groups. No randomization protocol was used. No blinding protocol was used.

## Data availability

The source data of this paper are collected in the following database record: biostudies:S-SCDT-10_1038-S44319-025-00406-8.

## Peer review information

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

## Acknowledgements

The work was funded by: "Regenerative Mechanisms for Health" project MAB/2017/2 carried out within the International Research Agendas program of the Foundation for Polish Science co-financed by the European Union under the European Regional Development Fund; National Science Centre, Poland (2019/35/B/NZ3/04066); (2019/33/B/NZ5/00780); (2019/35/B/NZ3/04095). Klaudia Maruszczak is supported by EMBO long-term fellowship - European

Molecular Biology Organization (ALTF 388-2023). We are grateful to Anna Marusiak, Maciej Cieśla, Minji Kim, and Vanessa Linke for scientific discussion.

## Author contributions

**Ben Hur Marins Mussulini**: Formal analysis; Validation; Investigation; Methodology; Writing—original draft; Writing—review and editing. **Klaudia K Maruszczak**: Formal analysis; Validation; Investigation; Methodology; Writing—original draft; Writing—review and editing. **Piotr Draczkowski**: Formal analysis; Validation; Investigation; Methodology; Writing—original draft; Writing—review and editing. **Mayra A Borrero-Landazabal**: Formal analysis; Validation; Investigation; Methodology; Writing—original draft; Writing—review and editing. **Selvaraj Ayyamperumal**: Formal analysis; Validation; Investigation; Methodology; Writing—original draft; Writing—review and editing. **Artur Wnorowski**: Formal analysis; Validation; Investigation; Methodology; Writing—original draft; Writing—review and editing. **Michal Wasilewski**: Validation; Investigation; Methodology; Writing—original draft; Writing—review and editing. **Agnieszka Chacinska**: Conceptualization; Supervision; Funding acquisition; Writing—original draft; Project administration; Writing—review and editing.

Source data underlying figure panels in this paper may have individual authorship assigned. Where available, figure panel/source data authorship is listed in the following database record: biostudies:S-SCDT-10_1038-S44319-025-00406-8.

## Disclosure and competing interests statement

The authors declare no competing interests.

# Expanded View Figures

**Figure EV1. Complex I accessory subunits KO cells screen for metabolic profile and resistance to AIFM1-induced cell death.**

(A) control HEK293T cells and correspondent complex I accessory subunits knock outs (KO) cell lines viability accessed by The CellTiter 96® AQueous One Solution Reagent after cell death induced by AIFM1. Data shown are mean ± SEM ($n = 3$ biological replicates) Statistical significance was obtained by an ordinary one-way ANOVA with Bonferroni multiple comparisons test (compared to HEK293T: NDUFA7-KO, $p > 0.9999$; NDUFS6-KO, $p > 0.9999$; NDUFV3-KO, $p = 0.0003$; NDUFA13-KO, $p = 0.0012$; NDUFA5-KO, $p = 0.0153$; NDUFS5-KO, $p > 0.9999$; NDUFB4-KO, $p = 0.0064$; NDUFB7-KO, $p > 0.9999$). *$p < 0.05$ indicates a significant difference compared to the HEK293T cells. (B) control HEK293T cells and correspondent complex I accessory subunits KO cell lines survival accessed by sulforhodamine b after cell death induced by AIFM1. Data shown are mean ± SEM ($n = 3$ biological replicates). Statistical significance was obtained by an ordinary one-way ANOVA with Bonferroni multiple comparisons test (compared to HEK293T: NDUFA7-KO, $p > 0.9999$; NDUFS6-KO, $p > 0.9999$; NDUFV3-KO, $p = 0.0002$; NDUFA13-KO, $p = 0.0002$; NDUFA5-KO, $p = 0.0489$; NDUFS5-KO, $p > 0.9999$; NDUFB4-KO, $p = 0.0314$; NDUFB7-KO, $p > 0.9999$). *$p < 0.05$ indicates a significant difference compared to the HEK293T cells. (C) Complex I activity of the control HEK293T cells and correspondent complex I accessory subunits KO cell lines. Data shown are mean ± SEM ($n = 3$ biological replicates). Statistical significance was obtained by an ordinary one-way ANOVA with Bonferroni multiple comparisons test (compared to HEK293T: NDUFA7-KO, $p < 0.0001$; NDUFS6-KO, $p < 0.0001$; NDUFV3-KO, $p < 0.0001$; NDUFA13-KO, $p < 0.0001$; NDUFA5-KO, $p < 0.0001$; NDUFS5-KO, $p < 0.0001$; NDUFB4-KO, $p < 0.0001$; NDUFB7-KO, $p < 0.0001$). *$p < 0.05$ indicates a significant difference compared to the HEK293T cells. (D) NADH/NAD$^+$ balance of the control HEK293T cells and correspondent complex I accessory subunits KO cell lines. Data shown are mean ± SEM ($n = 3$ biological replicates). Statistical significance was obtained by an ordinary one-way ANOVA with Bonferroni multiple comparisons test (compared to HEK293T: NDUFA7-KO, $p = 0.5571$; NDUFS6-KO, $p = 0.0476$; NDUFV3-KO, $p = 0.0007$; NDUFA13-KO, $p = 0.0018$; NDUFA5-KO, $p = 0.9997$; NDUFS5-KO, $p = 0.7218$; NDUFB4-KO, $p = 0.8921$; NDUFB7-KO, $p = 0.6560$). *$p < 0.05$ indicates a significant difference compared to the HEK293T cells. (E) control HEK293T, NDUFA13- and NDUFB4-KO cell lines metabolic consumption of 31 different metabolites. Data shown are mean ± SEM ($n = 4$ or 3 biological replicates per substrate). Statistical significance was obtained by an ordinary two-way ANOVA with Bonferroni multiple comparisons test. For succinate oxidation, $p < 0.0001$. For fumaric acid oxidation, $p = 0.0207$. For pyruvic acid + L-malic acid 100 µM oxidation, $p = 0.0390$. # $p < 0.05$ indicates difference between NDUFA13-KO and NDFUB4-KO. The remaining statistical comparisons are presented in Table EV3. (F) Cellular protein extracts were isolated from the control HEK293T cells and correspondent complex I accessory subunits KO cell lines. The samples were analyzed by reducing SDS–PAGE and Western blot. (G) Quantification of AIFM1 expression from (F) using ImageJ. Data shown are mean ± SEM ($n = 3$ biological replicates). Statistical significance was obtained by an ordinary one-way ANOVA with Bonferroni multiple comparisons test (compared to HEK293T: NDUFV3-KO, $p = 0.0015$; NDUFA5-KO, $p > 0.9999$; NDUFA13-KO; $p < 0.0001$; NDUFB4-KO; $p > 0.9999$). *$p < 0.001$ indicates a significant difference compared to the HEK293T cells. (H) Localization of mitochondrial proteins analyzed by limited degradation by proteinase K in intact mitochondria (250 mM sucrose) or mitoplasts (100, 25, and 5 mM sucrose). The samples were analyzed by SDS–PAGE and Western blot. OM, outer membrane; IM, inner membrane; IMS, intermembrane space.

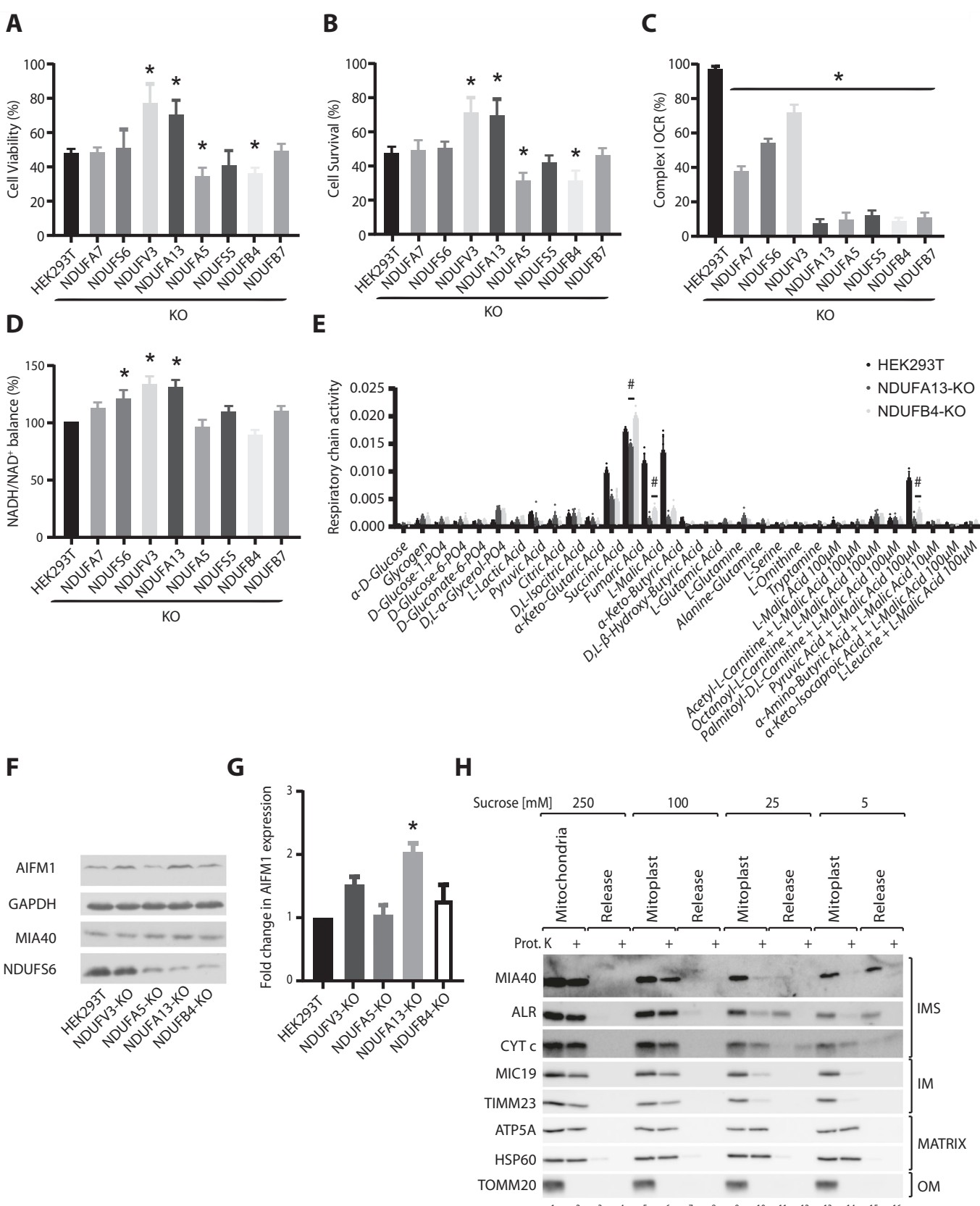

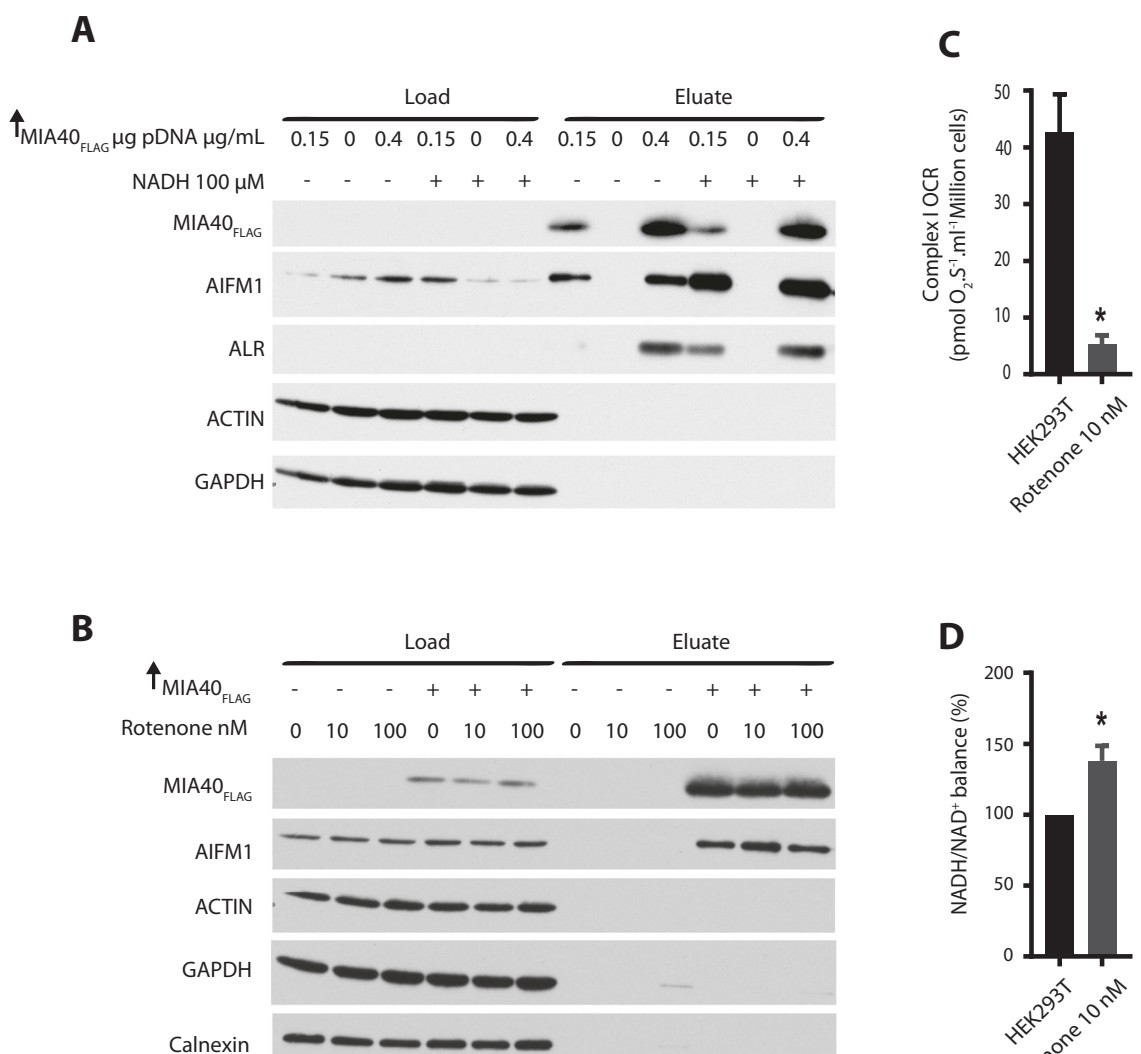

**Figure EV2. NADH increases AIFM1 and MIA40 interaction.**

(A) control HEK293T cells transfected with an empty plasmid or MIA40$_{FLAG}$ were solubilized, and the affinity purification of MIA40$_{FLAG}$ was performed in the presence of 100 μM NADH. Fractions were analyzed by SDS–PAGE and Western blot. Load 2.5%; Eluate: 100%. (B) HEK293T cells were transfected with an empty plasmid or MIA40$_{FLAG}$ for 48 h and then were incubated with 10 nM or 100 nM rotenone for 12 h. Afterwards, the cells were solubilized and the affinity purification of MIA40$_{FLAG}$ was performed. Fractions were analyzed by SDS–PAGE and Western blot. Load 2.5%; Eluate: 100%. (C) HEK293T cells complex I activity after 12 h treatment of 10 nM rotenone. Data shown are mean ± SEM ($n = 3$ biological replicates). Statistical significance was obtained by two-tailed unpaired t-test $p = 0.0007$. *$p < 0.05$ indicates significant differences between groups. (D) HEK293T cells were treated with 10 nM rotenone for 12 h and afterwards, the NADH/NAD$^+$ balance was measured. Data shown are mean ± SEM ($n = 3$ biological replicates). Statistical significance was obtained by two-tailed unpaired t-test $p = 0.0034$. *$p < 0.05$ indicates significant differences between groups.

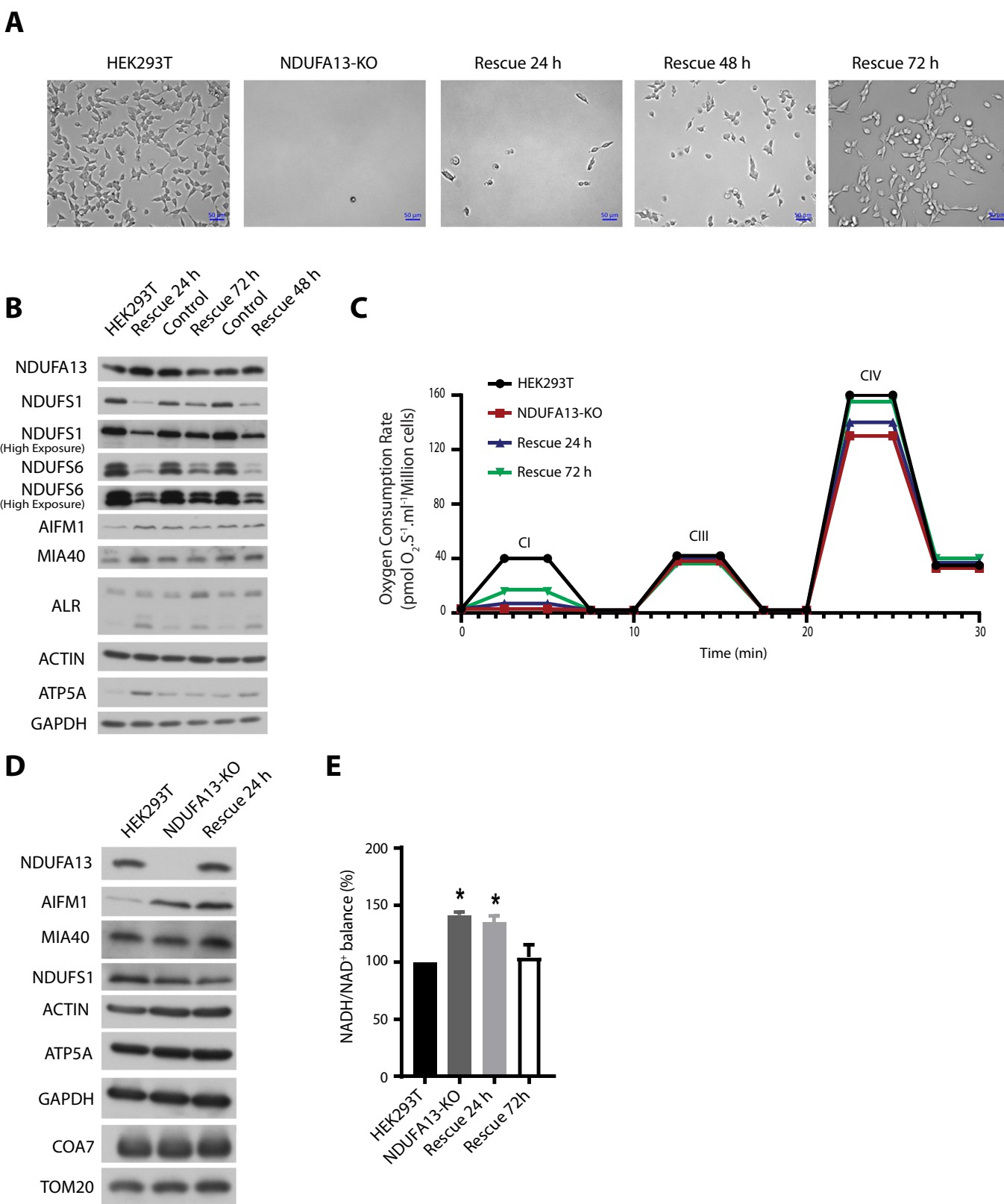

**Figure EV3.  Rescue of complex I activity in NDUFA13-KO cells.**

(A) Contrast microscopy capture of the control HEK293T cells, complex I accessory subunit NDUFA13-KO (NDUFA13KO), and overexpression of NDUFA13 by plasmid transfection in NDUFA13-KO cells after 24 h (Rescue 24 h), 48 h (Rescue 48 h), or 72 h (Rescue 72 h). Cells were cultured and transfected in high-glucose medium for indicated time points, and then shifted to galactose medium for 24 h. (B) Cellular protein extracts were isolated from the control HEK293T cells, Rescue 24, 48, and 72 h. Cells were cultured and transfected in high-glucose medium for indicated time points, and then shifted to galactose medium for 24 h. The samples were analyzed by reducing SDS–PAGE and Western blot. (C) High-resolution respirometry profile of control HEK293T, NDUFA13-KO, Rescue 24 and 72 h performed in high-glucose. (D) Cellular protein extracts were isolated from control HEK293T, NDUFA13-KO, and overexpression of NDUFA13 by plasmid transfection in NDUFA13-KO cells after 24 h in high-glucose. The samples were analyzed by reducing SDS–PAGE and Western blot. (E) NADH/NAD$^+$ balance was measured in the control HEK293T, NDUFA13-KO, Rescue 24 h and 72 h. Data shown are mean ± SEM ($n = 3$ biological replicates). Statistical significance was obtained by an ordinary one-way ANOVA with Bonferroni multiple comparisons test (compared to HEK293T: NDUFA13-KO, $p = 0.0001$; Rescue 24 h, $p = 0.0004$; Rescue 72 h, $p > 0.9999$). *$p < 0.05$ indicates a significant difference from HEK293T cells.

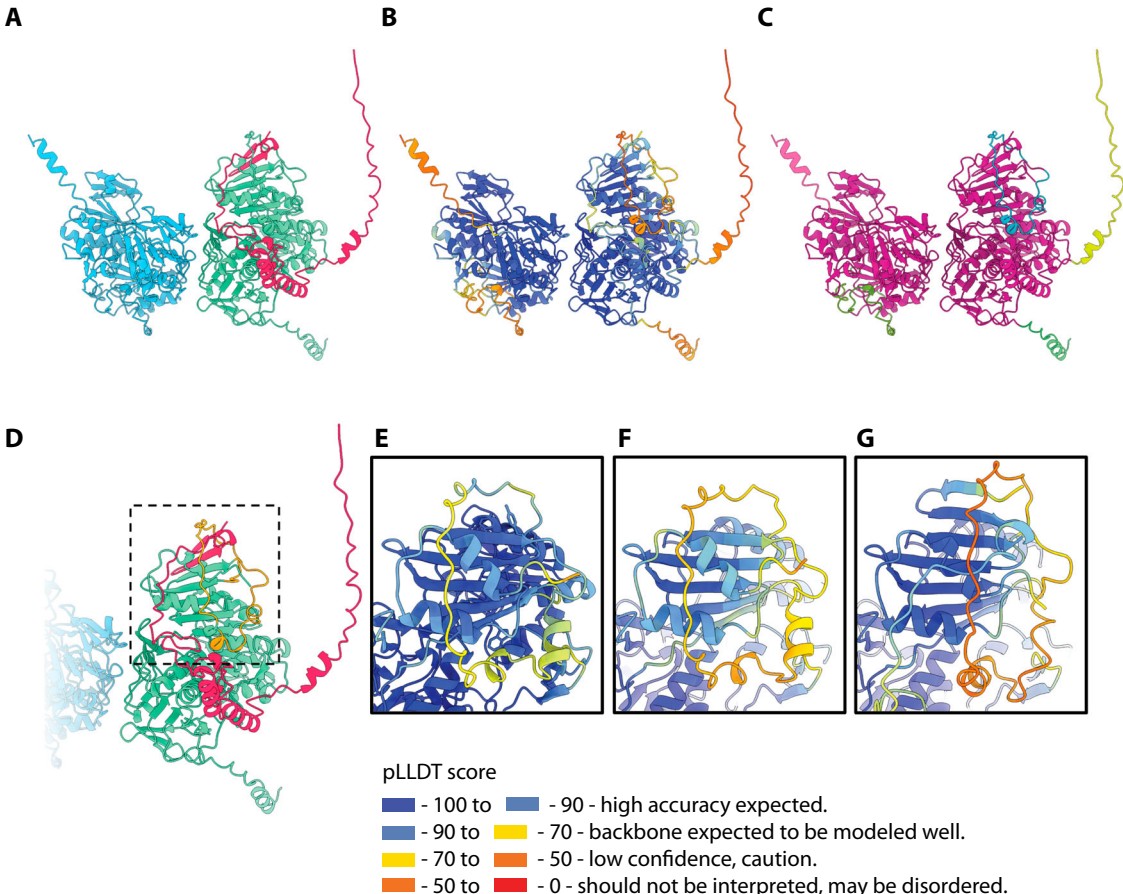

**pLLDT score**

- 100 to ☐ - 90 - high accuracy expected.
- 90 to ☐ - 70 - backbone expected to be modeled well.
- 70 to ☐ - 50 - low confidence, caution.
- 50 to ☐ - 0 - should not be interpreted, may be disordered.

**Figure EV4.   Accuracy of the predicted structure of the MIA40-AIFM1 dimer complex.**

The predicted structure of human MIA40-AIFM1 dimer complex is colored by (**A**), the chains composing the complex, (**B**) the AlphaFold per residue prediction confidence (pLLDT scores), (**C**) the Predicted Aligned Error (PAE), which reflects the confidence in the relative positions and orientations of parts of the predicted structure. Each color in (**C**) represents a coherent structural part of the complex, while the relative position of the parts highlighted by different colors remains ambiguous. (**D**) The overall structure of the MIA40-AIFM1 dimer with the region showed in the following panels indicated by the dashed line. The AlphaFold pLLDT scores for the AIFM1 C-loop showed a decrease in the confidence of the prediction when compared the (**E**), monomeric with (**F**), the dimeric form of the protein, indicating that the region is more likely to be unstructured when the AIFM1 dimerizes. (**G**) Further decrease of AIFM1 C-loop prediction quality was observed upon MIA40 binding.

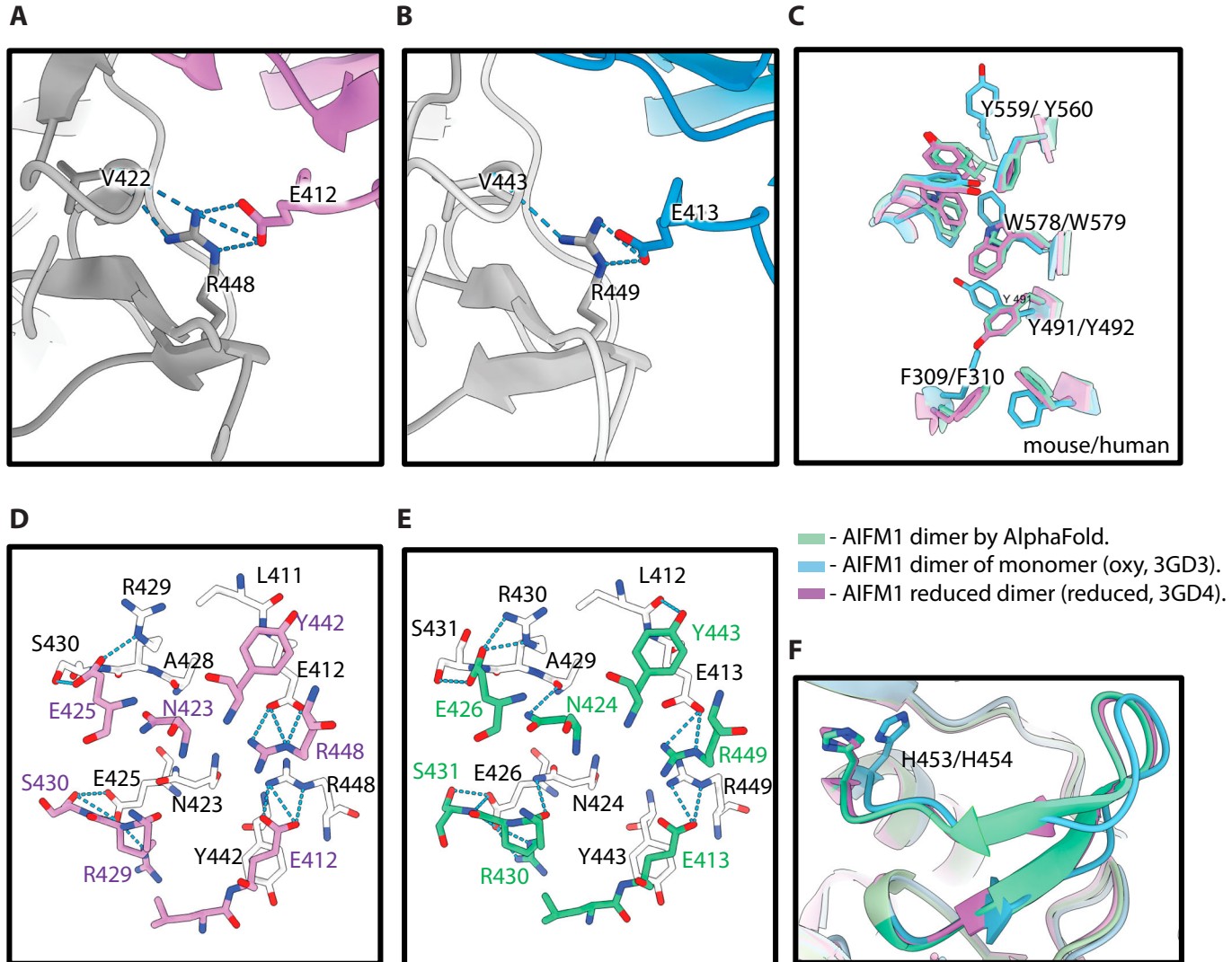

**Figure EV5. Comparison of the AIFM1 dimer predicted and experimentally determined structures of the AIFM1 dimer.**

(A) Hydrogen bond network formed by Glu412 and Arg448 in the naturally folded mouse dimer, and (B) the orthologous site formed by Glu413 and Arg449 in the predicted human AIFM1 dimer. (C) Side chains of residues that form the AIFM1 electron transfer chain analyzed in the crystal structures of mouse AIFM1 in oxidized monomeric (blue cartoon), its reduced dimeric form (pink cartoon), and the predicted human AIFM1 dimer (green cartoon). Residue numbers are provided for both orthologs of the protein (muse/human). The hydrogen bond network and conformation of the side chains at the dimerization interface were compared between (D), the naturally folded AIFM1 dimer, and (E), the predicted dimer. (F) The His454 (His453 in the mouse ortholog) side chain conformation was compared in the crystal structures of mouse oxidized monomeric AIFM1, its reduced dimeric form, and the predicted human AIFM1 dimer (color scheme same as in C).

**A**

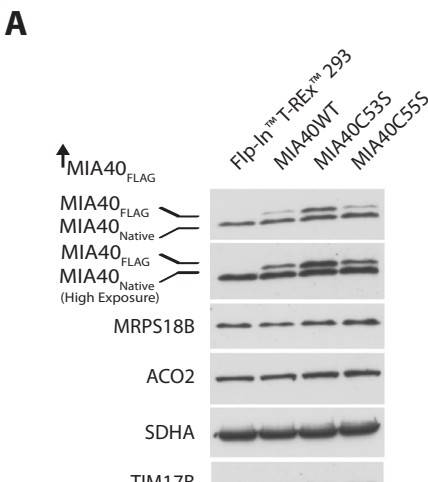

**B**

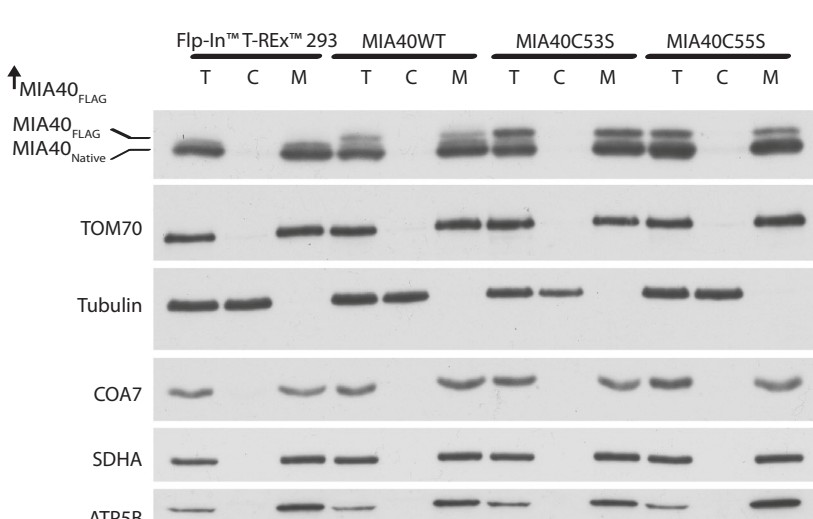

**Figure EV6. Protein levels and localization of MIA40 variants.**

(**A**) Total protein levels and (**B**) subcellular fractionation of Flp-In T-REx293 cells with induced expression of wild-type or MIA40_FLAG variants C53S or C55S (MIA40WT, MIA40C53S or MIA40C55S, respectively). Total post-nuclear supernatant (T), cytosol (C), and mitochondria (M). The samples were analyzed by SDS–PAGE and Western blot.

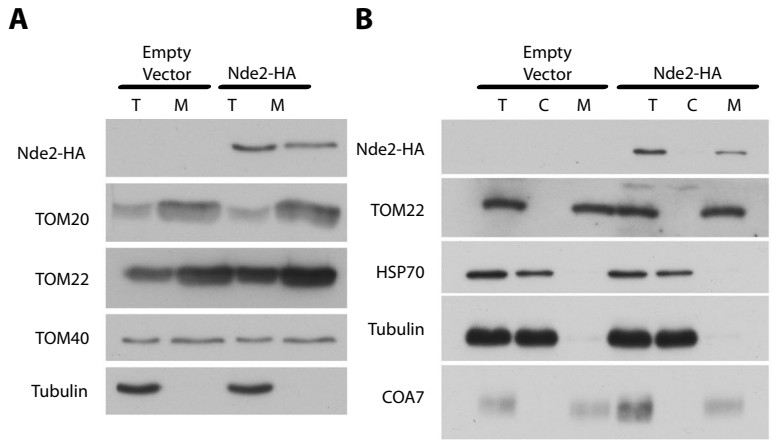

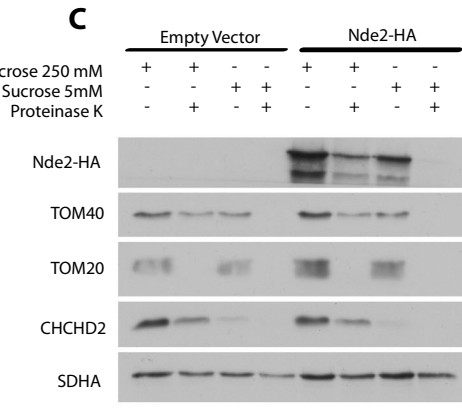

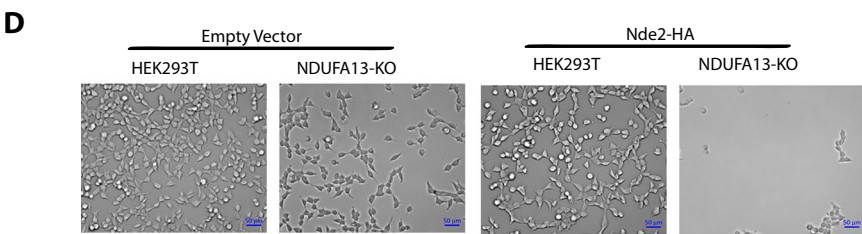

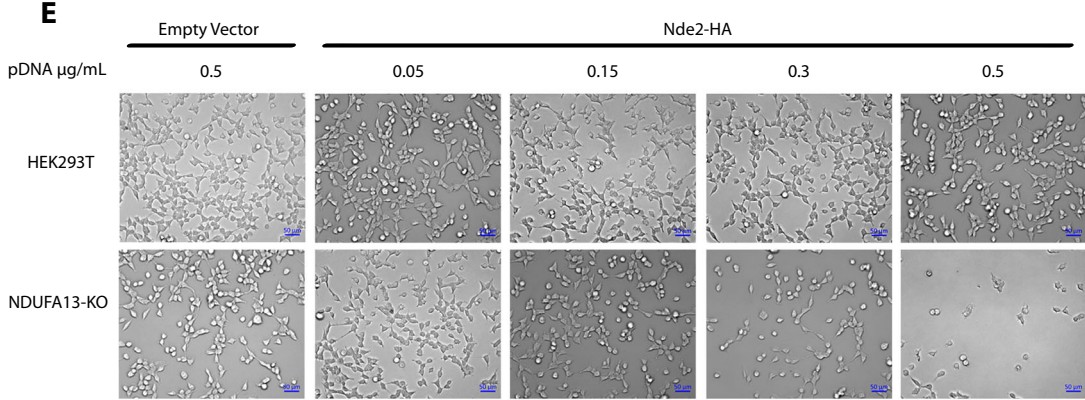

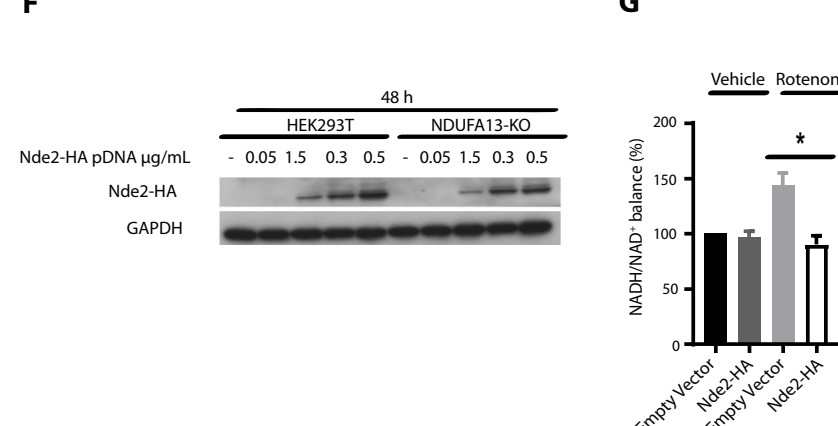

◄ **Figure EV7. Topology and activity of external NADH-ubiquinone oxidoreductase 2 tagged with HA at C-terminus (Nde2-HA).**

(A) Subcellular localization of Nde2-HA upon its transfection in the HEK293T cells. Cellular protein extracts (T) or isolated mitochondria (M) of HEK293T cells transfected with an empty plasmid or Nde2-HA were analyzed by reducing SDS–PAGE and Western blot. (B) subcellular fractionation of the HEK293T cells transfected with an empty vector or Nde2-HA. Total post-nuclear supernatant (T), cytosol (C), and mitochondria (M). The samples were analyzed by SDS–PAGE and Western blot. (C) Submitochondrial localization of Nde2-HA upon its transfection in the HEK293T cells analyzed by limited degradation by proteinase K in intact mitochondria (250 mM sucrose) or mitoplasts (5 mM sucrose). The samples were analyzed by SDS–PAGE and Western blot. (D) Contrast microscopy of HEK293T and complex I accessory subunit NDUFA13-KO transfected with an empty plasmid or Nde2-HA. (E) Contrast microscopy images of HEK293 and NDUFA13-KO cells transfected with increasing concentrations of the Nde2-HA plasmid after 48 h. (F) Cellular protein extracts were isolated from HEK293T and NDUFA13-KO cells that were transfected with different concentrations of Nde2-HA plasmid for 48 h. The samples were analyzed by reducing SDS–PAGE and Western blot. (G) HEK293T cells were transfected with an empty vector or Nde2-HA for 48 h. Then, the cells were incubated with a vehicle or 10 nM rotenone for 12 h. Afterward, the measurements of NADH/NAD$^+$ balance were performed. Data shown are mean ± SEM ($n = 3$ biological replicates). Statistical significance was obtained by an ordinary one-way ANOVA with Bonferroni multiple comparisons test (compared to HEK293T; HEK293T + Nde2-HA, $p > 0.9999$; HEK293T + rotenone, $p = 0.0003$; HEK293T + Nde2-HA + rotenone, $p = 0.4753$). *$p < 0.05$ indicates a significant difference from HEK293T cells treated with rotenone.

