## [Peer Review File · EMBO Reports]

MIA40 suppresses cell death induced by Apoptosis-inducing factor 1.

Ben Hur Mussulini, Klaudia Maruszczak, Piotr Draczkowski, Mayra Borrero Landazabal, Selvaraj Ayyamperumal, Artur Wnorowski, Michal Wasilewski, and Agnieszka Chacinska

Corresponding author(s): Agnieszka Chacinska (a.chacinska@imol.institute)

Review Timeline:

Submission Date:	25th Jun 24
Editorial Decision:	31st Jul 24
Revision Received:	13th Nov 24
Editorial Decision:	20th Dec 24
Revision Received:	8th Jan 25
Accepted:	24th Jan 25

Editor: Deniz Senyilmaz Tiebe

Transaction Report:

Dear Prof. Chacinska,

Thank you for submitting your research manuscript to our journal, which was now seen by three referees, whose reports are copied below. Attached are the two PDBs, which referee #3 ran in AlphaFold3 relevant to his/her point 1 in the review.

Referees express interest in the proposed role of MIA40 in AIF1 mediated cell death. However, they also raise significant concerns that need to be addressed to consider publication here.

I find the reports informed and constructive, and believe that addressing the concerns raised will significantly strengthen the manuscript. As the reports are below, and I think all points need to be addressed, I will not detail them here. Please contact me if you have questions or comments regarding the revision for further discussion(also by video chat).

Given these positive recommendations, we would like to invite you to submit a revised manuscript. Please revise your manuscript with the understanding that the referee concerns (as in their reports) must be fully addressed and their suggestions taken on board. Please address all referee concerns in a complete point-by-point response. Acceptance of the manuscript will depend on a positive outcome of a second round of review. It is EMBO reports policy to allow a single round of major experimental revision only and acceptance or rejection of the manuscript will therefore depend on the completeness of your responses included in the next, final version of the manuscript.

We realize that it is difficult to revise to a specific deadline. In the interest of protecting the conceptual advance provided by the work, we recommend a revision within 3 months. Please discuss the revision progress ahead of this time with me if you require more time to complete the revisions, or if you have questions or comments regarding the revision (also by video chat).

1. A data availability section providing access to data deposited in public databases is missing (where applicable).
2. Your manuscript contains statistics and error bars based on $n=2$. Please use scatter plots in these cases.

You can submit the revision either as a Scientific Report or as a Research Article. For Scientific Reports, the revised manuscript can contain up to 5 main figures and 5 Expanded View figures, and it should not exceed 27000 characters. If the revision leads to a manuscript with more than 5 main figures it will be published as a Research Article. In this case the Results and Discussion section should be separate. If a Scientific Report is submitted, these sections have to be combined. This will help to shorten the manuscript text by eliminating some redundancy that is inevitable when discussing the same experiments twice. In either case, all materials and methods should be included in the main manuscript file.

4) a .docx formatted letter INCLUDING the reviewers' reports and your detailed point-by-point responses to their comments. As part of the EMBO publication's Transparent Editorial Process, EMBO reports publishes online a Review Process File (RPF) to accompany accepted manuscripts. This File will be published in conjunction with your paper and will include the referee reports,

your point-by-point response and all pertinent correspondence relating to the manuscript.

<https://www.embopress.org/page/journal/14693178/authorguide#transparentprocess>

5) a complete author checklist, which you can download from our author guidelines

<https://www.embopress.org/page/journal/14693178/authorguide>. Please insert information in the checklist that is also reflected in the manuscript. The completed author checklist will also be part of the RPF.

6) Please note that all corresponding authors are required to supply an ORCID ID for their name upon submission of a revised manuscript (<<https://orcid.org/>>). Please find instructions on how to link your ORCID ID to your account in our manuscript tracking system in our Author guidelines

<<https://www.embopress.org/page/journal/14693178/authorguide#authorshipguidelines>>

7) Before submitting your revision, primary datasets produced in this study need to be deposited in an appropriate public database (see <https://www.embopress.org/page/journal/14693178/authorguide#datadeposition>). Please remember to provide a reviewer password if the datasets are not yet public. The accession numbers and database should be listed in a formal "Data Availability" section placed after Materials & Method (see also

<https://www.embopress.org/page/journal/14693178/authorguide#datadeposition>). Please note that the Data Availability Section is restricted to new primary data that are part of this study. * Note - All links should resolve to a page where the data can be accessed. *

Additional information on source data and instruction on how to label the files are available:

<https://www.embopress.org/page/journal/14693178/authorguide#sourcedata>

9) Our journal encourages inclusion of *data citations in the reference list* to directly cite datasets that were re-used and obtained from public databases. Data citations in the article text are distinct from normal bibliographical citations and should directly link to the database records from which the data can be accessed. In the main text, data citations are formatted as follows: "Data ref: Smith et al, 2001" or "Data ref: NCBI Sequence Read Archive PRJNA342805, 2017". In the Reference list, data citations must be labeled with "[DATASET]". A data reference must provide the database name, accession number/identifiers and a resolvable link to the landing page from which the data can be accessed at the end of the reference. Further instructions are available at <http://www.embopress.org/page/journal/14693178/authorguide#referencesformat>

10) Regarding data quantification (see Figure Legends:

<https://www.embopress.org/page/journal/14693178/authorguide#figureformat>)

11) The journal requires a statement specifying whether or not authors have competing interests (defined as all potential or actual interests that could be perceived to influence the presentation or interpretation of an article). In case of competing interests, this must be specified in your disclosure statement. Further information: <https://www.embopress.org/competing->

interests

12) Please also note our reference format:

13) All Materials and Methods need to be described in the main text using our 'Structured Methods' format, which is required for all research articles. According to this format, the Methods section includes a Reagents and Tools Table (listing key reagents, experimental models, software and relevant equipment and including their sources and relevant identifiers) followed by a Methods and Protocols section describing the methods using a step-by-step protocol format. The aim is to facilitate adoption of the methodologies across labs. More information on how to adhere to this format as well as a downloadable template (.docx) for the Reagents and Tools Table can be found in our author guidelines:

I look forward to seeing a revised version of your manuscript when it is ready. Please let me know if you have questions or comments regarding the revision.

Kind regards,

Deniz Senyilmaz Tiebe

Deniz Senyilmaz Tiebe, PhD
Senior Scientific Editor
EMBO Reports

Referee #1:

In this manuscript, Mussulini et al elegantly demonstrate, using a variety of approaches and cell models, that an increase in the NADH/NAD⁺ ratio strengthens the MIA40-AIFM1 interaction, resulting in a decrease in AIFM1-mediated cell death. However, the metabolic side of this model is a bit weak and should be strengthened by the authors.

Specific comments

From the published literature and the new data in this manuscript, the AIFM1-MIA40 complex seems to serve as a redox switch which measures metabolic conditions in the mitochondria and translates them into a binary life/death decision. Inhibiting mitochondrial OXPHOS via complex I disruption/inhibition results in phenomenal metabolic reprogramming. Depending on the cellular model/setting it can lead to an increase in glycolysis/glucose uptake, increased glutaminolysis and/or growth inhibition through induction of a metabolic shut-down and redox imbalance (to name several possible outcomes).

Which metabolic reprogramming occurs in each of the cellular models described in the Mussulini et al manuscript? The NADH/NAD⁺ ratio, measured by the authors using a commercial kit, is only one of the many metabolic changes that take place following complex I disruption/inhibition. NADH seems to be involved in regulating the formation of the AIFM1-MIA40 complex, but this is likely the tip of the iceberg. Thus, the authors are advised to perform metabolomics of the different cell models to obtain a metabolic landscape, that will help to decipher the binary life/death decision of this complex.

Referee #2:

In this work, Mussulini et al. found that MIA40 modulates cell death interacting with AIF1 (apoptosis-inducing factor), union that is dependent on NADH levels. They found that NDUFA13-KO cells show increased NADH levels that correlate with an augmented MIA1:AIF interaction and diminished cell death upon STS treatment. They also provided mechanistic information showing a putative interaction surface that involves the cysteine55 of MIA40. While this novel regulatory mechanism is interesting for the field and relevant as it may have a role in therapeutics to overcome cancer resistance to cell death, in my opinion, this hypothesis is not sufficiently supported by the evidence presented here. Thus, I consider that this work is not suitable for publication in EMBO Reports in its current form.

Major points:

1. Throughout the manuscript, the authors refer to AIFM1-induced cell death to describe staurosporine-induced cell death when caspases are inhibited by Z-VAD. However, it should be noted that some authors have reported that STS-dependent cell death in MEF cells is only partially related to AIF (Milastra et al., 2016, Immunity). Therefore, this terminology should be revised carefully for a more accurate representation of the treatment. Moreover, staurosporine induces a plethora of cellular activities that may or may not result in cell death.
2. Similarly, when comparing the different NDUFA accessory subunits (Extended Data 1), the authors correlate alterations in cell viability using MTS and sulforhodamine assays to differences in AIFM1-dependent cell death. However, alterations in cellular metabolism are not necessarily related to cell death, especially considering that different adaptor complex I subunits are knocked out and compared, which is crucial for cellular metabolism. Indeed, as author comment "we concluded that the degree of complex I impairment did not correlate with the sensitivity or resistance of AIFM1-induced cell death in these cells" (lines 115-116) and that the NDUFA13-KO cells (selected as best condition to represent NADH increased conditions) exhibit different doubling time compared to the wt cells (line 157-160). Thus, additional characterization of cell death induction in NDUFA KO cells by PI+/AnnexinV (as it is done in the rest of the paper) would be more illustrative.
3. High NADH levels seem to correlate with higher expression of AIFM1 and increased MIA40-FLAG_AIFM1 interaction; however, its implication in cell death is not very clearly described. In the rescue experiments where NDUFA13 is reintroduced, NDUFA13 protein shows total recovery after 24 hours and is maintained during 72 hours (Extended Data 3). Despite the authors reporting that resistance to death is related to an increased NADH/NAD balance, no data on NADH values are reported in the rescue experiments to support this hypothesis. This is key considering that NDUFA13 expression levels are wild-type-like after 24 hours, but overall cell death is not (Figure 2b)
4. Results with C55S mutant overexpression suggest that it is more toxic than MIA40 WT when stimulated with staurosporine, which correlates with a decreased MIA40-FLAG_AIFM1 interaction. At this point, it would be important to know if there is differential cell death induction in the absence of treatment, as well as to evaluate the expression levels and subcellular localization of the mutants, in order to support the idea that this residue is key for MIA40-FLAG_AIFM1 interaction surface. In this regard, a more detailed explanation of how the AlphaFold data supports the idea that this mutation disrupts or affects the interaction of these proteins is recommended, as most of the models have similar values for wt and C55S mutant.
5. Regarding the overexpression of Nde2-HA to reduce NADH, the authors chose a concentration that they visually reported to result in approximately 50% cell death (line 281, Fig. Ext. 8a). However, upon stimulation with staurosporine, they report around 25% cell death (Figure 5b). How are these scenarios compatible? Additionally, while Nde2-HA is reported to localize to the mitochondria, its overexpression seems to be more ubiquitous (compare the ratio Nde2-HA/TOM20 in Extended Figure 7b), suggesting that this effect could be more global and not specific to mitochondrial NADH.

Minor:

1. The term "control" is used throughout the paper but refers to different conditions, such as treated with staurosporine (e.g., Fig. 1C), untreated (e.g., Fig. 1E), or wild-type cells (e.g., Extended 3b). This inconsistency makes the figures difficult to interpret, in my opinion.
2. In Extended Figure 2b, I believe MIA40-FLAG expression is incorrectly labelled as it appears in every column (+).
3. In the silencing experiments (Figure 3g), the healthy KO population seems to be shifted compared to the WT and silencing conditions. Additionally, the number of represented cells appears to be very different.
4. Figure 2f is missing.

Referee #3:

Teasing the role of AIFM1 in complex I regulation and cell death is important and has been of a subject of debate but the exact mechanism of how AIFM1 is regulated in cell death remains poorly understood.

In manuscript "MIA40 modulates cell death induced by Apoptosis-inducing factor 1" Mussulini et al. describe a metabolic dependence of MIA40 inhibition of AIFM1-mediated cell death which is brought about by an increase in the NADH/NAD⁺ ratio. The authors screened a previously described library of complex I-deficient human cells to identify one (NDUFA13-KO) with an increase in NADH/NAD⁺ balance which they used to test the hypothesis that MIA40 modulates AIFM1 pro-death activity. They propose a direct interaction model based on AlphaFold multimer that supports the interaction of 1 MIA40 protein with the AIFM1 dimer and may explain the structural basis of the interaction considering the NADH vs NAD⁺ bound states of AIFM1. By using an overexpression and deletion or silencing strategies of the various components the authors were able to modulate the pro-death role of AIFM1 in MIA40 and NADH-dependent manner.

The study is interesting furthering the understanding of the MIA40-AIFM1 regulation even though the AlphaFold model has not

been probed functionally but it could be. I have a few suggestions to help improve the study.

1. I have done an alphafold3 run for the MIA40-AIFM1 complex (I attached it to the report) which also places the cofactors FAD and NAD in the corresponding binding sites. I wonder if the authors are also aware about the potential binding of an additional NADH (PDB ID 4BUR; <https://pubs.acs.org/doi/10.1021/bi500343r>). This may explain possibly how parts that are buried may be displaced to allow MIA40-AIFM1 interaction at higher NADH/NAD⁺ levels.
2. The authors say that they study parthanatos pg7 ln 149 but also AIFM1 pro-apoptotic activity pg 4 ln 87. So which is it. When caspase is inhibited with zVAD apoptosis is inhibited. Could you use pharmacologic inhibition of parthanatos to show that this is what may be causing the death. There is quite a bit of evidence suggesting that the conditions used to induce cell death may also induce necroptosis and a necroptosis inhibitor may also be useful to test.
3. Speaking of cell death, I find the assay description in the methods not very clear. Could you please clarify the assay description: "AIFM1 cell death was induced by z-vad-fmk 100 μ M pre-treatment during 30 min followed by staurosporine 2 μ M treatment during 3.5 h in presence of z-vad-fmk 100 μ M (Daugas et al., 2000). Medium was changed previous cell death induction. Cell viability and cell survival was accessed in 96 well plates after seeding 20x10³ cells during 24 h in presence of low glucose medium." I find these sentences confusing and it is important to carefully describe it given the scope of the study is to understand the role of MIA40-AIFM1 complex in cell death.

EMBOR-2024-59852V2

Mussulini et al.

Point-by-point response to the Reviewers' comments

We would like to thank all three reviewers for their constructive criticism. We have now addressed all the reviewer's comments. We improved several experiments and it resulted in new versions of the figures: 2a, 3g, 5a, EV3a, EV7d and 7Eve. Moreover, we did new experiments upon reviewers' requests and this resulted in new figures: 1a, 1b, 4a, 4b, EV1e, EV3e, EV6a, EV6b and EV7b. We believe that our effort improved greatly the quality of our manuscript and we hope that the reviewers will find our conclusions justified.

Reviewers' comments:

Referee #1: In this manuscript, Mussulini et al elegantly demonstrate, using a variety of approaches and cell models, that an increase in the NADH/NAD⁺ ratio strengthens the MIA40-AIFM1 interaction, resulting in a decrease in AIFM1-mediated cell death. However, the metabolic side of this model is a bit weak and should be strengthened by the authors.

Specific comments

From the published literature and the new data in this manuscript, the AIFM1-MIA40 complex seems to serve as a redox switch which measures metabolic conditions in the mitochondria and translates them into a binary life/death decision. Inhibiting mitochondrial OXPHOS via complex I disruption/inhibition results in phenomenal metabolic reprogramming. Depending on the cellular model/setting it can lead to an increase in glycolysis/glucose uptake, increased glutaminolysis and/or growth inhibition through induction of a metabolic shut-down and redox imbalance (to name several possible outcomes).

Which metabolic reprogramming occurs in each of the cellular models described in the Mussulini et al manuscript? The NADH/NAD⁺ ratio, measured by the authors using a commercial kit, is only one of the many metabolic changes that take place following complex I disruption/inhibition. NADH seems to be involved in regulating the formation of the AIFM1-MIA40 complex, but this is likely the tip of the iceberg. Thus, the authors are advised to perform metabolomics of the different cell models to obtain a metabolic landscape, that will help to decipher the binary life/death decision of this complex.

To answer the reviewer's comment, we performed metabolic profiling of the control HEK293T, NDUFA13-KO and NDUFB4-KO cells using MITOPLATE S-I, a technique that allows comprehensive profiling of mitochondrial function (Kuzniewska et al., 2020; Mosejová

et al., 2021; Radogna et al., 2021; Verberk et al., 2022). For this experiment, we chose NDUFA13-KO and NDUFB4-KO, the cell lines that proved to be the most resistant and the most sensitive to cell death, respectively. Additionally, both cell lines were marked by the loss of complex I activity. The results of this experiment are presented in the new Figure EV1e. Even though both, NDUFA13-KO and NDUFB4-KO, showed drastic decrease in the activity of complex I, they exhibited distinct metabolic profiles, suggesting different adaptive strategies. Therefore, in the case of NDUFA13-KO cells, increased NADH/NAD⁺ balance is a result of a particular adaptive strategy, which we decided to elucidate in our manuscript.

Reviewer #2: In this work, Mussulini et al. found that MIA40 modulates cell death interacting with AIF1 (apoptosis-inducing factor), a function that is dependent on NADH levels. They found that NDUFA13-KO cells show increased NADH levels that correlate with an augmented MIA1:AIF interaction and diminished cell death upon STS treatment. They also provided mechanistic information showing a putative interaction surface that involves the cysteine55 of MIA40. While this novel regulatory mechanism is interesting for the field and relevant as it may have a role in therapeutics to overcome cancer resistance to cell death, in my opinion, this hypothesis is not sufficiently supported by the evidence presented here. Thus, I consider that this work is not suitable for publication in EMBO Reports in its current form.

Major points:

1. Throughout the manuscript, the authors refer to AIFM1-induced cell death to describe staurosporine-induced cell death when caspases are inhibited by Z-VAD. However, it should be noted that some authors have reported that STS-dependent cell death in MEF cells is only partially related to AIF (Milastra et al., 2016, Immunity). Therefore, this terminology should be revised carefully for a more accurate representation of the treatment. Moreover, staurosporine induces a plethora of cellular activities that may or may not result in cell death.

To further confirm that cell death phenotypes observed in this study depend on AIFM1 induction, we took advantage of two previously described inhibitors of AIFM1-induced cell death, DPQ and Necrostatin-1 (Park et al., 2014). AIFM1-induced cell death is mediated by RIP1-dependent activation of PARP1 (Jouan-Lanhouet et al., 2012; Xu et al., 2007). Necrostatin-1 and DPQ are inhibitors of RIP1 and PARP-1, respectively, and they were reported before to drastically reduce the presence of AIFM1 in the cytosol after the caspase-independent cell death induction (Park et al., 2014). In our study, both inhibitors were able to prevent more than 80% of AIFM1-induced cell death in cells expressing MIA40C55S, confirming that we are observing the AIFM1-related cell death. The results are presented in the new Figure 4A and 4B.

2. Similarly, when comparing the different NDUF accessory subunits (Extended Data 1), the authors correlate alterations in cell viability using MTS and sulforhodamine assays to differences in AIFM1-dependent cell death. However, alterations in cellular metabolism are not necessarily related to cell death, especially considering that different adaptor complex I subunits are knocked out and compared, which is crucial for cellular metabolism. Indeed, as author comment "we concluded that the degree of complex I impairment did not correlate with the sensitivity or resistance of AIFM1-induced cell death in these cells" (lines 115-116) and that the NDUFA13-KO cells (selected as best condition to represent NADH increased conditions) exhibit different doubling time compared to the wt cells (line157-160). Thus, additional characterization of cell death induction in NDUF KO cells (as it is done in the rest of the paper) would be more illustrative.

We followed the recommendations of the reviewer and characterized cell death phenotypes by Pi⁺/AnnexinV in all KO cell lines used in this study. The new results are presented in the Figure 1A and 1B. We obtained similar results as in our previous experiments using MTS and Sulfohodamine B. The new cell death profiles obtained from flow cytometry confirm that the NDUFA13-KO and NDUV3-KO cells are more resistant to AIFM1-induced cell death, while the NDUFA5-KO and NDUFB4-KO cells are more sensitive to AIFM1-induced cell death.

3. High NADH levels seem to correlate with higher expression of AIFM1 and increased MIA40-FLAG_AIFM1 interaction; however, its implication in cell death is not very clearly described. In the rescue experiments where NDUFA13 is reintroduced, NDUFA13 protein shows total recovery after 24 hours and is maintained during 72 hours (Extended Data 3). Despite the authors reporting that resistance to death is related to an increased NADH/NAD balance, no data on NADH values are reported in the rescue experiments to support this hypothesis. This is key considering that NDUFA13 expression levels are wild-type-like after 24 hours, but overall cell death is not (Figure 2b).

Following the reviewer's suggestion, we measured the NADH/NAD⁺ balance after 24 h and 72 h of NDUFA13 recovery. The results are shown in the new figure EV3e. Even though the levels of NDUFA13 were recovered after 24 hours, the NADH/NAD⁺ balance was still increased. However, after 72 hours, not only the levels of NDUFA13, complex I assembly and activity were partially recovered, the NADH/NAD⁺ balance returned to the levels of the control.

4. Results with C55S mutant overexpression suggest that it is more toxic than MIA40 WT when stimulated with staurosporine, which correlates with a decreased MIA40-FLAG_AIFM1 interaction. At this point, it would be important to know if there is differential cell death induction in the absence of treatment, as well as to evaluate the expression levels and subcellular localization of the mutants, in order to support the idea that this residue is key for MIA40-FLAG_AIFM1 interaction surface. In this regard, a more detailed explanation of how the AlphaFold data supports the idea that this mutation disrupts or affects the interaction of these proteins is recommended, as most of the models have similar values for wt and C55S mutant.

Addressing the first part of the reviewer's comment, we checked the AIFM1-induced cell death of Flp-In T-Rex 293 control cells and Flp-In T-Rex 293 that overexpressed wild-type MIA40 and MIA40 variants, C53S and C55S, that were treated with a vehicle. We could not observe any differences in cell death phenotypes in all the tested cells (Figure 1 of this

report).

Figure 1 AIFM1-induced cell death of Flp-IN T-Rex 293 cells, and cells overexpressing wild-type and MIA40 C53S and C55S mutants, that were treated with a vehicle.

Next, we showed that the cellular levels of wild-type and MIA40C55S were relatively equal in the total fraction (new figure EV6a) and both variants, C53S and C55S, were localized in mitochondria (new figure EV6b). Lastly, we showed that the interaction between MIA40C55S and AIFM1 was significantly reduced as compared to MIA40 wild-type (Figure 4c). Our AlfaFold2 (AF2) and AlphaFold3 (AF3) results collectively indicated that there might be a loss of interaction between residue R239 of the second AIFM1 of the dimer and the MIA40 residues D32 and P36 upon the C55S substitution. This could be one of the possible explanations for why MIA40C55S reduces the interaction between MIA40 and AIFM1 dimer. However, the AF2 and AF3 models did not consistently predict this interaction. The AF2 predicted loss of hydrogen bond between the AIFM1 R239 and D32 of MIA40C55S accompanied by the reduction of AIFM1 R239 contacts with MIA40C55S P36 when compared with the complex MIA40WT, while AF3 indicated loss of hydrogen bond between the AIFM1 R239 and P36 of MIA40C55S or no effect of the MIA40 mutation at all. This variability in the predicted interaction might be related to the fact that both involved

interaction sites, AIFM1 R239 and MIA40C55S D32-P36, are located in long unstructured loops (according to the AF2/AF3 predicted structure) and are far from C55S. This predicted interaction site still awaits experimental confirmation, and therefore, we do not want to comment further as it would be too speculative. Our manuscript is based on solid physiological data, and thus, we do not want to dilute their relevance. To milder the conclusions drawn based on AlphaFold, we added to our manuscript “Despite these supportive arguments, only subsequent structural studies on MIA40 and AIFM1 dimer will be able to address this question”.

5. Regarding the overexpression of Nde2-HA to reduce NADH, the authors chose a concentration that they visually reported to result in approximately 50% cell death (line 281, Fig. Ext. 8a). However, upon stimulation with staurosporine, they report around 25% cell death (Figure 5b). How are these scenarios compatible? Additionally, while Nde2-HA is reported to localize to the mitochondria, its overexpression seems to be more ubiquitous (compare the ratio Nde2-HA/TOM20 in Extended Figure 7b), suggesting that this effect could be more global and not specific to mitochondrial NADH.

Firstly, we wanted to have a situation that after the transfection of Nde2-HA, 50% of the NDUFA13-KO cells would remain alive. Afterwards, we induced cell death in the remaining living cells, which probably was a confusing part. We included the following correction in the text “*Next, we tested whether the modulation of NADH balance would reverse the phenotype of cell death observed in the NDUFA13-KO cells. The Nde2-HA expression was adjusted to result in approximately 50% cell death of NDUFA13-KO cells (Fig EV7e). Both the HEK293T cells and the remaining living NDUFA13-KO cells expressed similar levels of Nde2-HA after 48 h of transfection (Fig EV7f). Next, we induced cell death via AIFM1 in the remaining living NDUFA13-KO cells after Nde2-HA expression*”. In regard to Nde2-HA localization, we performed a new subcellular fractionation with the total, cytosolic and mitochondrial fractions. The results are presented in the new figure EV7b. We showed that Nde2-HA localizes to the mitochondrial fraction. We did not detect Nde2-HA in the cytosol, therefore we are certain that the described events are the results of the activity of Nde2 in mitochondria.

Minor.

1. The term "control" is used throughout the paper but refers to different conditions, such as treated with staurosporine (e.g., Fig. 1C), untreated (e.g., Fig. 1E), or wild-type cells (e.g., Extended 3b). This inconsistency makes the figures difficult to interpret, in my opinion.

We answered the reviewer's comment by specifying which control was used in a given experiment.

2. In Extended Figure 2b, I believe MIA40-FLAG expression is incorrectly labelled as it appears in every column (+).

The figure was corrected.

3. In the silencing experiments (Figure 3g), the healthy KO population seems to be shifted compared to the WT and silencing conditions. Additionally, the number of represented cells appears to be very different.

We agree with the reviewer's comment. The gating of the wild-type sample was not optimal in this run, hence we replicated this experiment and included a new representative Figure 3g.

4. Figure 2f is missing.

This mistake was fixed.

Reviewer #3: Teasing the role of AIFM1 in complex I regulation and cell death is important and has been of a subject of debate but the exact mechanism of how AIFM1 is regulated in cell death remains poorly understood.

In manuscript "MIA40 modulates cell death induced by Apoptosis-inducing factor 1" Mussulini et al. describe a metabolic dependence of MIA40 inhibition of AIFM1-mediated cell death which is brought about by an increase in the NADH/NAD⁺ ratio. The authors screened a previously described library of complex I-deficient human cells to identify one (NDUFA13-KO) with an increase in NADH/NAD⁺ balance which they used to test the hypothesis that MIA40 modulates AIFM1 pro-death activity. They propose a direct interaction model based on AlphaFold multimer that supports the interaction of 1 MIA40 protein with the AIFM1 dimer and may explain the structural basis of the interaction considering the NADH vs NAD⁺ bound states of AIFM1. By using an overexpression and deletion or silencing strategies of the various components the authors were able to modulate the pro-death role of AIFM1 in MIA40 and NADH-dependent manner.

The study is interesting furthering the understanding of the MIA40-AIFM1 regulation even though the AlphaFold model has not been probed functionally but it could be. I have a few suggestions to help improve the study.

1. I have done an alphafold3 run for the MIA40-AIFM1 complex (I attached it to the report) which also places the cofactors FAD and NAD in the corresponding binding sites. I wonder if the authors are also aware about the potential binding of an additional NADH (PDB ID 4BUR; <https://pubs.acs.org/doi/10.1021/bi500343r>). This may explain possibly how parts that are buried may be displaced to allow MIA40-AIFM1 interaction at higher NADH/NAD⁺ levels.

We thank the Reviewer for his effort in generating the AlphaFold3 (AF3) models. Indeed, the AF3 models suggested that an additional cofactor molecule occupying the second NAD binding site, as proposed by Ferreira et al. (Biochemistry 2014) in the PDB 4BUR structure, would lead to the displacement of the AIFM1 C-loop as the two severely clash in the predicted model. In addition to the models provided by the Reviewer; we generated 5 additional AF3 models (each with a different seed) of the MIA40-AIFM1 complex with one molecule of FAD and NAD per AIFM1 protomer and 5 more models of the complex with one FAD and two NAD per AIFM1 protomer. The predicted structures consistently showed that the position of the C-loop, docked against the rest of AIFM1, and the NAD molecule occupying the second NAD-binding pocket are mutually exclusive as they result in severe clashes (Fig. R2 A). The same can be observed when comparing the PDB 4BUR structure reported by Ferreira et al. with the atomic models of oxidized and reduced form of naturally folded murine AIFM1 (PDB 3GD3 and 3GD4) published by Sevrioukova (JMB 2009). In the models deposited by Sevrioukova, the C-loop was determined only in the oxidized AIFM1 (PDB 3GD4), where it was docked against the AIFM1 in a position that would hinder the binding of the second NAD molecule. Indeed, the C-loop was released and remained too dynamic to be resolved in the structure of the reduced dimeric AIFM1 (PDB 3GD4), even though the second NAD-binding site was not occupied by the cofactor in this structure. Although it is pleasing to speculate, in the light of the AF3 results, that the higher NADH/NAD⁺ levels could displace the C-loop to facilitate the MIA40-AIFM1 interaction, it has to be noted that the prediction confidence of the second NAD molecule and the surrounding region were relatively low, rendering the structural details of the binding rather speculative (Fig. R2 B). Moreover, we also observed several discrepancies between our AF2 and AF3 models in the other parts of the complex (Fig. R3). Although the discrepancies did not affect our overall biological conclusions drawn from the AF2 structures, they would have to be addressed in the manuscript upon inclusion of the AF3 results. Our manuscript is mainly based on physiological data, and we are afraid that extending the discussion by comparing the differences between two versions of the AlphaFold would divert the attention of a reader.

Fig. R2. AlphaFold3 prediction of the second NAD molecule binding by the MIA40-AIFM1 complex. (A) Comparison of the model provided by the Reviewer and one of the top-ranked models calculated by the authors of the study. (B) AlphaFold3 model coloured by the per-residue prediction confidence scores (pLDDT) showing the uncertainty of the predicted binding mode of the second NAD molecule.

Fig. R3. Discrepancies in the structural details of the atomic models predicted by AlphaFold2 and AlphaFold3.

2. The authors say that they study parthanatos pg7 In 149 but also AIFM1 pro-apoptotic activity pg 4 In 87. So which is it. When caspase is inhibited with zVAD apoptosis is inhibited. Could you use pharmacologic inhibition of parthanatos to show that this is what may be causing the death. There is quite a bit of evidence suggesting that the conditions used to induce cell death may also induce necroptosis and a necroptosis inhibitor may also be useful to test.

To validate that cell death phenotypes observed in this study are related to AIFM1, we used two inhibitors of the latter, namely DPQ and Necrostatin-1. We described the experiments in the answer to reviewer 2. (major point 1.) of this report.

We replaced dubious nomenclature by “AIFM1 induced cell death” or “cell death via AIFM1” in the new version of the manuscript.

3. Speaking of cell death, I find the assay description in the methods not very clear. Could you please clarify the assay description: "AIFM1 cell death was induced by z-vad-fmk 100 μ M pre-treatment during 30 min followed by staurosporine 2 μ M treatment during 3.5 h in presence of z-vad-fmk 100 μ M(Daugas et al., 2000). Medium was changed previous cell death induction. Cell viability and cell survival was accessed in 96 well plates after seeding 20×10^3 cells during 24 h in presence of low glucose medium." I find these sentences confusing and it is important to carefully describe it given the scope of the study is to understand the role of MIA40-AIFM1 complex in cell death.

We corrected the method section that describes cell death induction, as follows:

For viability and survival experiments, 2×10^4 cells of each cell line were plated in 96 well plates. On the following day medium was changed to avoid false positive cell death events, and cells were pre-treated with 100 μ M of z-vad-fmk, the pan-caspase inhibitor, for 30 min. Next, cells were stimulated with 2 μ M staurosporine for 3.5 h to induce cell death. The z-vad-fmk 100 μ M was kept in the medium during the total 4 h treatment (Daugas et al., 2000). Cell viability was accessed by CellTiter 96® AqueousOne Solution Cell Proliferation Assay (Promega Corporation), following manufactory instructions, reaction time lasted 1h. Sulforhodamine B colorimetric assay for cytotoxicity was used to access cell survival accordingly to previous publication (Vichai and Kirtikara, 2006). Cell death was assessed by annexinV (AV) and propidium iodide (PI) staining using Alexa Flour™ 488 AnnexinV/Dead cell apoptosis kit (Thermo Fisher Scientific) following manufacturer instructions. Data was acquired using a BD LSR Fortessa or Agilent NovoCyte flow cytometers to measure percentages of Annexin V and PI positive cells to compare cell death among different HEK293T and the other complex I accessory subunit KOs used in this study (Fig 1a). Data was analyzed using FlowJo or Novo Express software as described previously (Cieřla et al., 2021). For the confirmation of cell death phenotype related to AIFM1, cells were pretreated with Necrostatin-1 60 μ M for 1 h or DPQ 30 μ M for 14 h. All data was normalized to the respective vehicle control (DMSO 0.2%).

References

- Cieřla, M., Ngoc, P. C. T., Cordero, E., Martinez, Á. S., Morsing, M., Muthukumar, S., Beneventi, G., Madej, M., Munita, R., Jönsson, T., Lövgren, K., Ebbesson, A., Nodin, B., Hedenfalk, I., Jirström, K., Vallon-Christersson, J., Honeth, G., Staaf, J., Incarnato, D., . . . Bellodi, C. (2021). Oncogenic translation directs spliceosome dynamics revealing an integral role for SF3A3 in breast cancer. *Molecular Cell*, 81(7), 1453-1468.e12. <https://doi.org/10.1016/j.molcel.2021.01.034>
- Daugas, E., Susin, S.A., Zamzami, N., Ferri, K.F., Irinopoulou, T., Larochette, N., Prévost, M., Leber, B., Andrews, D., Penninger, J., Kroemer, G., 2000. Mitochondrio-nuclear translocation of AIF in apoptosis and necrosis. *FASEB j.* 14, 729–739. <https://doi.org/10.1096/fasebj.14.5.729>
- Fagnani, E., Cocomazzi, P., Pellegrino, S., Tedeschi, G., Scalvini, F.G., Cossu, F., Da Vela, S., Aliverti, A., Mastrangelo, E., Milani, M., 2024. CHCHD4 binding affects the active site of apoptosis inducing factor (AIF): Structural determinants for allosteric regulation. *Structure* 32, 594-602.e4. <https://doi.org/10.1016/j.str.2024.02.008>
- Ferreira, P., Villanueva, R., Martínez-Júlvez, M., Herguedas, B., Marcuello, C., Fernandez-Silva, P., Cabon, L., Hermoso, J.A., Lostao, A., Susin, S.A., Medina, M., 2014. Structural Insights into the Coenzyme Mediated Monomer–Dimer Transition of the Pro-Apoptotic Apoptosis Inducing Factor. *Biochemistry* 53, 4204–4215. <https://doi.org/10.1021/bi500343r>
- Jouan-Lanhouet, S., Arshad, M.I., Piquet-Pellorce, C., Martin-Chouly, C., Le Moigne-Muller, G., Van Herreweghe, F., Takahashi, N., Sergent, O., Lagadic-Gossmann, D., Vandenabeele, P., Samson, M., Dimanche-Boitrel, M.-T., 2012. TRAIL induces necroptosis involving RIPK1/RIPK3-dependent PARP-1 activation. *Cell Death Differ* 19, 2003–2014. <https://doi.org/10.1038/cdd.2012.90>
- Kuzniewska, B., Cysewski, D., Wasilewski, M., Sakowska, P., Milek, J., Kulinski, T.M., Winiarski, M., Kozielwicz, P., Knapska, E., Dadlez, M., Chacinska, A., Dziembowski, A., Dziembowska, M., 2020. Mitochondrial protein biogenesis in the synapse is supported by local translation. *EMBO Reports* 21, e48882. <https://doi.org/10.15252/embr.201948882>
- Mosejová, E., Bosnjakovic, R., Kubala, L., Vaříček, O., 2021. Pseurotin D Induces Apoptosis through Targeting Redox Sensitive Pathways in Human Lymphoid Leukemia Cells. *Antioxidants* 10, 1576. <https://doi.org/10.3390/antiox10101576>
- Park, E.J., Min, K., Lee, T.-J., Yoo, Y.H., Kim, Y.-S., Kwon, T.K., 2014. β -Lapachone induces programmed necrosis through the RIP1-PARP-AIF-dependent pathway in human hepatocellular carcinoma SK-Hep1 cells. *Cell Death Dis* 5, e1230–e1230. <https://doi.org/10.1038/cddis.2014.202>
- Radogna, F., Gérard, D., Dicato, M., Diederich, M., 2021. Assessment of Mitochondrial Cell Metabolism by Respiratory Chain Electron Flow Assays, in: Weissig, V., Edeas, M. (Eds.), *Mitochondrial Medicine, Methods in Molecular Biology*. Springer US, New York, NY, pp. 129–141. https://doi.org/10.1007/978-1-0716-1266-8_9
- Sevrioukova, I. F. (2009). Redox-Linked conformational dynamics in Apoptosis-Inducing factor. *Journal of Molecular Biology*, 390(5), 924–938. <https://doi.org/10.1016/j.jmb.2009.05.013>
- Verberk, S.G.S., de Goede, K.E., Gorki, F.S., van Dierendonck, X.A.M.H., Argüello, R.J., Van den Bossche, J., 2022. An integrated toolbox to profile macrophage immunometabolism. *Cell Reports Methods* 2, 100192. <https://doi.org/10.1016/j.crmeth.2022.100192>

- Vichai, V., & Kirtikara, K. (2006). Sulforhodamine B colorimetric assay for cytotoxicity screening. *Nature Protocols*, 1(3), 1112–1116. <https://doi.org/10.1038/nprot.2006.179>
- Xu, X., Chua, C.C., Kong, J., Kostrzewa, R.M., Kumaraguru, U., Hamdy, R.C., Chua, B.H.L., 2007. Necrostatin-1 protects against glutamate-induced glutathione depletion and caspase-independent cell death in HT-22 cells. *Journal of Neurochemistry* 103, 2004–2014. <https://doi.org/10.1111/j.1471-4159.2007.04884.x>

Dear Agnieszka,

Thank you for submitting your revised manuscript. It has now been seen by two of the original referees.

As you can see, both referees find that the study is significantly improved during revision and recommend publication. However, I need you to address the points below before I can accept the manuscript.

- Please address the remaining minor concerns of referees #2 and #3.
- Please rename the Conflicts of Interest section as Disclosure Statement and Competing Interests.
- Please remove the Author Contributions section from the manuscript.
- As per our format requirements, in the reference list, citations should be listed in alphabetical order and then chronologically, with the authors' surnames and initials inverted; where there are more than 10 authors on a paper, 10 will be listed, followed by 'et al.'. Please see <https://www.embopress.org/page/journal/14693178/authorguide#referencesformat>
- We note the following about the figure callouts: Fig 4e is currently not called out in the text. There is a callout for Extended Data Fig. 8b, which was not submitted.
- The tables titled Table - Dataset EV1 and Table - Dataset EV2 are better suited to be Expanded View Tables. They should be resubmitted as Table EV1 and Table EV2. Please update their source file names, titles in the manuscript tracking system, figure legends in the manuscript, callouts in the manuscript.
- Reagents & Tools table needs to be removed from the manuscript text and resubmitted as a separate file.
- Papers published in EMBO Reports include a 'synopsis' and 'bullet points' to further enhance discoverability. Both are displayed on the html version of the paper and are freely accessible to all readers. The synopsis includes a short standfirst summarizing the study in 1 or 2 sentences (max 35 words) that summarize the paper and are provided by the authors and streamlined by the handling editor. I would therefore ask you to include your synopsis blurb and 3-5 bullet points listing the key experimental findings.
- We note that the synopsis image was submitted in PDF format. Please resubmit it in jpeg, TIFF or png format. In addition, please adjust the size to fit our format requirements - i.e. 550 (width) x 300-600 (height) pixels. Lastly, please increase the font sizes of the labels to increase their readability.
- Our production/data editors have asked you to clarify several points in the figure legends:
 - o Please define the annotated p values * as well as provide the exact p-values for the same in the legend of figure EV 1d; as appropriate.
 - o Please note that the exact p values are not provided in the legends of figures 1b, e; 2b, d; 3h; 4b, d; 5b, d; EV 1a-c, e, g; EV 2c-d; EV 3e; EV 7g.
 - o Although 'n' is provided, please describe the nature of entity for 'n' in the legends of figures 1b, e; 2b, d; 3h; 4b, d; 5b, d; EV 1a-c, e, g; EV 2c-d; EV 3e; EV 7g.

Thank you again for giving us to consider your manuscript for EMBO Reports, I look forward to your minor revision.

Kind regards,

Deniz

--

Deniz Senyilmaz Tiebe, PhD
Senior Scientific Editor
EMBO Reports

Referee #2:

One minor aspect related to previous point 1:

Staurosporine treatment has been reported to induce various types of regulated cell death. While inhibiting RIP1 and PARP reduces cell death, this does not necessarily indicate a direct connection to AIFM1-mediated cell death. Although both proteins are involved in the AIFM1 pathway, they also participate in other regulated cell death mechanisms.

Moreover, the reduction in cell death observed with these inhibitors, from approximately 37-40% to 10-15%, suggests their involvement in other pathways. For example, necrostatin, which inhibits necroptosis-a form of cell death that staurosporine has also been reported to trigger when caspases are inhibited (PMCID: PMC3409216)-and PARP cleavage, is associated with classical apoptosis, another process triggered by staurosporine (PMID: 11423986).

Therefore, it is recommended to refer to the phenomenon as "staurosporine-induced cell death" or consider the possibility that not all cell death observed in your experiments is exclusively linked to AIFM1.

Referee #3:

The reference Fagnani et al., 2024 should be added to the reference list.

Interesting that in addressing the comments by 2 reviewers the authors downplayed alphafold insights as not being physiological, yet they expanded the interpretation of the models significantly in the results section.

All editorial and formatting issues were resolved by the authors.

Prof. Agnieszka Chacinska
IMol Polish Academy of Sciences
Warsaw
Poland

Dear Agnieszka,

Thank you for submitting your revised manuscript. I have now looked at everything and all is fine. Therefore, I am very pleased to accept your manuscript for publication in EMBO Reports.

Congratulations on a nice work!

Before we can transfer your manuscript to our production, I need your input on one more point. I suggest some minor alterations in the items below to increase clarity and accessibility of the findings. Please take a look and confirm, or feel free to propose further changes.

Title: MIA40 suppresses cell death induced by Apoptosis-inducing factor 1.

Abstract: Mitochondria harbor respiratory complexes that perform oxidative phosphorylation. Complex I is the first enzyme of the respiratory chain that oxidizes NADH. A dysfunction in complex I can result in higher cellular levels of NADH, which in turn strengthens the interaction between apoptosis-inducing factor 1 (AIFM1) and Mitochondrial intermembrane space import and assembly protein 40 (MIA40) in the mitochondrial intermembrane space. We investigated whether MIA40 modulates the activity of AIFM1 upon increased NADH/NAD⁺ balance. We found that in model cells characterized by an increase in NADH the AIFM1-MIA40 interaction is strengthened and these cells demonstrate resistance to AIFM1-induced cell death. Either silencing of MIA40, rescue of complex I, or depletion of NADH through the expression of yeast NADH-ubiquinone oxidoreductase-2 sensitized NDUFA13-KO cells to AIFM1-induced cell death. These findings indicate that the complex of MIA40 and AIFM1 suppresses AIFM1-induced cell death in a NADH-dependent manner. This study identifies an effector complex involved in regulating the programmed cell death that accommodates the metabolic changes in the cell and provides a molecular explanation for AIFM1-mediated chemoresistance of cancer cells.

Synopsis blurb

An increase in the NADH/NAD⁺ ratio due to Complex I defects results in strengthening of the MIA40 and AIFM1 dimer interaction in the intermembrane space of mitochondria and resistance to AIFM1-induced cell death.

Bullet points

1. Complex I dysfunction as well as other metabolic alterations lead to an increased NADH/NAD⁺ ratio in the intermembrane space of mitochondria.
2. High NADH/NAD⁺ ratio strengthens the interaction between MIA40 and the AIFM1 dimer which renders the cells resistant to cell death.
3. Decrease in NADH/NAD⁺ ratio weakens the interactions between MIA40 and AIFM1 and results in cell death sensitivity.
4. A decrease in MIA40 activity sensitizes the complex I defective cells to cell death.

Kind regards,

Deniz

--

Deniz Senyilmaz Tiebe, PhD
Senior Scientific Editor
EMBO Reports

--
